



# Using OMPS-LP color ratio to extract stratospheric aerosol particle size and concentration with application to volcanic eruptions

Yi Wang[1], Mark Schoeberl[1], Ghassan Taha[2], Daniel Zawada[3], Adam Bourassa[3]

[1]Science and Technology Corporation (STC), Columbia, MD, United States
[2]Morgan State University, Baltimore, MD, United States
[3]Institute of Space and Atmospheric Studies, University of Saskatchewan, Saskatoon, Saskatchewan, Canada

*Correspondence to*: Yi Wang (yiwang@stcnet.com)

**Abstract.** We develop an algorithm that uses the aerosol extinction at two wavelengths (color ratio) to derive the size and number density for stratospheric aerosols. We apply our algorithm to Ozone Mapping Profiler Suite Limb Profiler (OMPS-
LP) L2 and Stratospheric Aerosol and Gas Experiment (SAGE) data. We show that the color ratio between two wavelengths (e.g. 510 nm/869 nm) is insensitive to aerosol concentration and thus can be used to derive aerosol size assuming a log-normal size distribution. With the size and the extinction, we can compute a number density consistent with both wavelengths. Our results compare favorably to balloon borne particle size and concentration measurements. Our results are also consistent with SAGE solar occultation measurements. Finally, we show the background distribution of stratospheric aerosols and the changes
in those distributions during the Reikoke and Hunga Tonga-Hunga Ha'apai volcanic eruptions. We also show the evolution of the size and number density of aerosols following both of those eruptions.

## 1 Introduction

Stratospheric aerosols are of significant interest to the scientific community because of their impact of the climate through volcanic and pyroCb injections (e.g. Turco et al, 1982; Robock, 2000, Kremser et al., 2016), and because of the possibility of
altering their abundance for geoengineering purposes (Robock, 2014; NASEM, 2021). The concentration of stratospheric aerosols (particles in the size range from 0.01 to 1 µm radius) varies from year to year due to sporadic injection events, and these events have been connected to short term changes in climate (Kremser et al., 2016).

Stratospheric aerosols mostly consist of small sulfuric acid droplets that form in the cold, high relative humidity lower
stratosphere after the oxidation of sulphur compounds. Volcanic ash, PyroCB smoke and dust also contribute to the aerosol abundance in the lower stratosphere. Ash and dust mostly consist of larger particles that can settle out of the stratosphere within a few days. Smoke, dust and ash can provide condensation nuclei for sulfuric acid, forming dark aerosols that have been observed to 'self loft' due to solar heating (Yu et al., 2019, Khaykin et al., 2020, Khaykin et al., 2022).



Stratospheric aerosol remote sensing started with solar occultation systems.  For example, the family of Stratospheric Aerosol Measurement (SAM) and Stratospheric Aerosol and Gas Experiment (SAGE) satellite series, combined with other solar occultation satellites has provided global monitoring of the stratospheric aerosol layer since 1975 (McCormick et al., 1982). The solar occultation technique directly measures the aerosol optical depth at multiple wavelengths, usually 300 – 1000 nm, and thus can estimate the column average aerosol concentration and size (Thomason and Taha, 2003).


Newer approaches to satellite stratospheric aerosol measurements include lidar methods (e.g. Cloud-Aerosol LiDAR and Infrared Pathfinder Satellite Observations (CALIPSO), Vernier et al., 2009) and limb scattering methods used by the Ozone Mapping Profiler Suite Limb Profiler (OMPS-LP) (Jaross et al., 2014) as well as the Optical Spectrograph and InfraRed Imaging System (OSIRIS) (Llewellyn et al., 2004).  The limb scattering method provides improved global coverage. For

example, SAGE III on the International Space Station (SAGE III/ISS) occultation system takes about a month to provide near-global coverage, whereas OMPS-LP can provide equivalent coverage in a day.  On the other hand, the limb scattering radiances are more complicated to interpret and have higher uncertainty due to variations in the scattering angle with orbit, variation in scattering properties with particle size, and measurement contamination by surface reflected light.  Yet, despite these drawbacks, the limb scattering techniques have provided new insights into the rapid evolution of stratospheric aerosols

especially immediately following volcanic eruptions (Taha et al., 2022, Khaykin et al., 2022, Wells et al., 2022).

The goal of this paper is to provide a new approach to derive particle size distribution from OMPS-LP measurements.  OMPS-LP is on the Suomi National Polar-orbiting Partnership (S-NPP) satellite and is described in Flynn et al. (2014).  A copy of the S-NPP instrument has recently been launched on NOAA-21.  The current OMPS-LP aerosol algorithm (Taha et al. 2021)

computes the extinction at each OMPS measured wavelength using a fixed aerosol size distribution model.  A radiative transfer model then computes the scattered radiance using a first-guess for the aerosol extinction, the assumed size distribution and solar scattering angle. The algorithm then iterates the aerosol concentration until the model computed radiance matches the observed radiance. In our algorithm, we use extinction at two wavelengths to constrain both the derived number density and size using the Level 2 (L2) extinction products from the current OMPS algorithm. The algorithm is based on Bourassa et al.

(2007) approach using OSIRIS measurements and Thomason and Vernier's (2013) work using SAGE II measurements that have been extended to SAGE III/ISS (Schoeberl et al., 2021; Kovilakam et al., 2022).  In the next section, we detail the algorithm and results. We then validate the algorithm by comparing our results to *in situ* balloon observations and SAGE III/ISS data.  Finally, we apply the algorithm to the background stratospheric aerosol distribution and two recent volcanic eruptions where aerosol extinctions were anomalous.



## 2 Aerosol particle size and concentration retrieval algorithm

Our goal is to derive the aerosol particle size and number density from extinction measurements. The algorithm uses the aerosol extinction (AE) coefficient from the L2 OMPS-LP products, V2.1 (Taha et al., 2021), which provides AE retrievals at multiple wavelengths. We use V2.1 AE values at two different wavelengths to calculate the color ratio (CR), which is defined here as AE ($\lambda_1$)/AE ($\lambda_2$) where $\lambda$ is the wavelength and $\lambda_1 < \lambda_2$. In the V2.1 L2 algorithm, AE is retrieved independently at each wavelength (Taha et al., 2021) using the Chahine nonlinear relaxation method (Chahine, 1970) and the Gauss–Seidel limb scattering (GSLS) radiative transfer model (Loughman et al., 2015). The OMPS-LP AE products have a vertical resolution of 1.6-1.8 km, reported every 1 km. Only data from the central slit profiler is used in this study.

To show how using the CR provides more information on the aerosol concentration and size, we start with a series of idealized extinction calculations. For these calculations, we simulate the Mie scattering process using the SASKTRAN radiative transfer model (Bourassa et al., 2008a; Zawada et al., 2015). SASKTRAN assumes a log-normal aerosol size distribution with a mode width of 1.6 for spherical sulphate aerosols. This size distribution is consistent with *in situ* stratospheric aerosol measurements (Deshler 2003; Bourassa 2014). In the discussion below, the 'size' is the median radius of the size distribution.

The aerosol extinction is equal to the number density ($N$) multiplied by the extinction cross-section ($\sigma$),

$$AE = \sigma \cdot N \tag{1}$$

In computing the color ratio of aerosol extinction (CR), the number density cancels out. Thus, the CR should be nearly independent of the aerosol concentration and only a function of size. Figure 1a shows the CR as a function of aerosol particle size computed from SASKTRAN's Mie code for two CRs 510nm/869nm and 745nm/869nm. Overall CR decreases with increasing particle size, and as the particle size increases to 0.4 µm, the CR approaches one. This CR – size relationship allows us to infer the median aerosol particle radius up to ~0.4 µm. Any two distinct wavelengths can be chosen for CR, but as the two wavelengths approach each other Fig. 1a shows that the CR gradient decreases and thus the uncertainty in the retrieved size increases. On the other hand, if one of the wavelengths is too short (< ~400 nm) Rayleigh scatter overwhelms the aerosol scattering signal at lower altitudes, and the aerosol extinction can't be measured. Figure 1b shows results of SASKTRAN simulations comparing CR (510nm / 869nm) and extinction coefficient values for both aerosol and ice particles. The ice particle model is from Baum et al. (2014). The particle size ranges are from 0.001 µm to 1.0 µm for aerosols, and 10 µm to 50 µm for ice particles, respectively. The number density is from 0.1 cm$^{-3}$ to 10000 cm$^{-3}$ for aerosols. For ice particles, different number densities have very similar CR values that are all ~ 1, so only the number density 0.01 cm$^{-3}$ is shown. This figure shows that the particle size is only a function of CR and is independent of the number density. To reiterate our basic algorithm approach, given the CR, we can compute size and the cross section. Once effective particle size is estimated, we can use the measured extinction at one of the wavelengths (e.g., either 869 nm or 510nm) and then use Eq. (1) to compute the aerosol number density. This method produces both consistent number density and particle size.

As noted above, the OMPS-LP L2 AE retrievals assume an aerosol model with fixed particle size. The L2 algorithm iterates
the modelled radiance to match the observed radiance. However, in the radiative transfer simulations, if we change to a smaller
particle size, a higher number density will produce the same observed radiance. In other words, the radiance measurement at
a single wavelength is insufficient to constrain both the number density and the particle size, thus the L2 algorithm produces
a different number density for each wavelength. However, AE is the robust quantity retrieved by the L2 V2.1 algorithm since
the AE must match the observed radiance. Thus, if we use the L2 AE at two wavelengths, we have enough information to
independently compute a size and number density consistent with the two AE values and independent of the radiative transfer
model assumptions about the L2 size distribution. This approach was also used by Bourassa et al (2008b).

One possible source of error in our calculation is the variation in Mie scattering phase function with size. In the V2.1 L2
algorithm, the Mie phase function variation is included in the radiative transfer model computation of the radiance. In our
algorithm, size can vary, and changes in the particle size may thus be inconsistent with the Mie scattering phase function.
Rieger et al., (2018) compared retrievals between volcanically quiescent periods (small aerosols only) and post eruptions
periods characterized by bimodal particle distributions. They did find a retrieval dependence on scattering angles; however,
the scattering angle error was minor when averaged over similar range of scattering angles. Since we will compare the
retrievals within the same latitudes during the same times, we believe that the biases caused by size-driven Mie phase function
error will be small.

Clouds produce very high extinctions, and to reduce possible cloud contamination, clouds are removed from the extinction
profile before retrieval. The L2 algorithm retrieves the cloud top height using a radiance gradient scheme (Chen et al., 2016).
Large aerosol particles with CR values of ~1 are uncommon in the stratosphere because these particles settle out within days.
More common is low-extinction sub-visible cirrus (SVC) seen near the tropopause (Schoeberl et al., 2021). We select CR limit
of < 1.1 to filter out cirrus that may appear above the L2 cloud top height. At the same time, this restriction limits the retrieved
aerosol sizes to < 0.3 μm. Figure 2 displays a cross sectional CR profile for a single OMPS-LP granule on 09/13/2020. The
magenta line in the figure is the cloud top height from the OMPS-LP file. The figure shows that CR of ~ 1.1 is close to the
retrieved cloud top height.

**3 Validation**

In this section we compare our OMPS-LP aerosol retrieved size and number density with *in situ* balloon measurements and
SAGE III/ISS satellite observations.





## 3.1 Comparison of OMPS-LP retrievals with Wyoming in situ balloon measurements

Balloon *in situ* stratospheric aerosol measurements is routinely made using the University of Wyoming Laser Particle Counter
(LPC) (Ward et al., 2014; Deshler et al. 2022). The LPC data is fit to a bimodal lognormal size distribution and the data files
report the parameters to these fits. To minimize the effect of spatial and temporal differences in aerosol distributions, we use
the OMPS-LP profiles having the closest time and location to balloon measurements. Figs. 3 and 4 show two characteristic
profiles. The particle size and number density retrieved from the OMPS-LP profile are shown as red lines, the black lines are
balloon observations with an uncertainty range. The uncertainty ranges of OMPS-LP retrievals are calculated from the
extinction coefficients (AE), using the formula below (2).

$$\frac{AE_1}{AE_2} \times (1 \pm \sqrt{(\frac{\delta_1}{AE_1})^2 + (\frac{\delta_2}{AE_2})^2})$$ (2)

where $\delta$ is the *AE* error provided in L2 files. The subscripts denote wavelengths, 510 nm and 869 nm, respectively.

The comparison of our OMPS-LP retrievals to Wyoming *in situ* balloon data shows good agreement in the lower stratosphere
where size distribution is not bimodal. In general, particle size between OMPS-LP retrievals and from balloon data are ~0.1
µm at all altitudes. The particle size decreases with increasing altitude in both OMPS-LP retrievals and in the balloon data
above 20 km. This distribution can be explained by the fact that smaller particles will have slower settling speeds and thus a
longer residence time in the stratosphere, whereas the larger particles will settle out to lower altitudes. In Fig 4, the region
where the balloon measured particle size distribution is strongly bimodal our retrieval shows a jump in particle size, which is
not unexpected. The color ratio method does not give us enough information to generate parameters for a bimodal distribution,
so the algorithm simply increases the size. The increase in size produces a decrease in number density as the extinction is
mostly due to fewer large particles (see Section 2). Based on these comparisons, we conclude that our algorithm does good
job reproducing the balloon measurements.

## 3.2 Comparison of OMPS-LP retrievals with SAGE III/ISS retrievals

SAGE III/ISS and predecessor instruments provide multi-wavelength extinction profiles from solar occultation measurements.
The advantage of SAGE measurements is that the extinction is measured directly through the attenuation of the solar beam.
SAGE is also 'self-calibrating' in that an exo-atmospheric measurement is made for each vertical scan. Taha et al. (2021)
compared coincidence and zonal mean V2.1 OMPS aerosol extinction with SAGE III/ISS and found good agreement at
equatorial latitudes but larger relative differences above 25 km outside of the tropics. For example, at high altitudes and
latitudes in the northern hemisphere, SAGE III compared to OMPS shows ~ 40% lower extinction for the shorter wavelengths
(Taha et al., 2021, Fig. 6). This means that the SAGE CR values will be smaller and particles relatively larger compared to
OMPS (Fig. 1b). Assuming the same extinction, SAGE will thus report a lower concentration compared to OMPS. With these
caveats in mind, using the SAGE color ratio/ extinction provides a test of our retrieval scheme and allows us to compare the
more complex extinction measurement made by OMPS-LP with SAGE III direct measurements.




Note that OMPS-LP makes measurements near 1:30 PM local time while SAGE III/ISS measurements are at local sunrise and sunset, thus for our comparisons, we follow Taha et al. (2021), and use the zonal median retrieval values for both from OMPS-LP and SAGE III/ISS. For the SAGE data, we use multiple profiles in the latitude range indicated, while OMPS-LP measurements are selected to be near coincident with the location of the SAGE profiles. We apply our algorithm to SAGE

III/ISS data but use a CR of 521 nm / 869 nm which are the available SAGE III/ISS extinction wavelengths close to OMPS. Aside from comparing the retrieved size and number density, we can also look at the influence of different OMPS-LP scattering angles in the retrievals. In Fig. 5 retrieved size and number density are compared for four different scattering angle regimes. In the figures, the measurement bands extend from the first quartile to the third quartile of the data, with a line at the zonal median. Some values of the SAGE III/ISS aerosol extinction coefficient are zero or negative due to the very low aerosol

concentrations during the observing period (Taha et al., 2021). This produces an artificially wide measurement uncertainty band.

In general, Fig. 5 shows that the retrievals from OMPS-LP and from SAGE III/ISS are closely matched at all altitudes. The typical particle size is in the range 0.1 μm and 0.15 μm and does not show significant variations between the two retrievals.

The number density from OMPS-LP is slightly larger than that from SAGE III/ISS as expected from the discussion above. There is no significant difference among each scattering angle range, except at the scattering angle of 33° above 24 km, where OMPS-LP has a lower particle size and higher number density. This result is consistent with Taha et al. (2021) comparisons where poorer agreement between OMPS-LP L2 retrievals and SAGE was found above 25 km. The agreement validates our assertion that errors due to Mie phase function variation with size are minor and that the extinction estimates from the OMPS-

LP L2 algorithm are robust. We also note that bias between SAGE and OMPS-LP in the northern extra tropics is consistent with Taha et al. (2021) comparison between OMPS-LP and SAGE extinction variations with wavelength.

## 4 Results from Background Aerosols and Volcanic Perturbations

As an application of our retrieval scheme, we show the variations in aerosol size and number density retrieved during background and two volcanic events. Recall that our retrieval cannot provide particle size information above 0.3μm, - larger

particles are identified as clouds and are removed.

### 4.1 Background

Figure 6a shows the distribution of CR vs AE measurements for background (non-volcanic) conditions. The distribution of the algorithm retrieved size and number densities are shown in Fig. 6b. The designated regions in Fig. 6a, c, e as aerosol and

sub-visible cirrus are based on the categorization in Schoeberl et al. (2021). Figure 6a shows a high concentration of measurements with extinctions of ~ $4 \times 10^{-4}$ km$^{-1}$ with CR values of 2-3. The region of lower extinction and higher CR around



this region may indicate aerosol formation or evaporation of small particles. The high concentration region connects to an SVC region with lower CR and higher extinction. The connection is due to larger particle formation.

Figure 6b is consistent with the interpretation above and results from the validation section. The highest concentration of aerosols observations is in the ~0.1 µm range with number density between 2 and 20 particles/cm$^3$. The screening of CR < 1.1 has reduced the number of high extinction particles in the SVC domain.

## 4.2 Aerosol changes during Reikoke volcano eruption

Reikoke (48.29 °N, 153.25 °W) erupted on June 21, 2019, and injected materials to a height of 17-19 km (Muser et al., 2020;
Gorkavyi et al., 2021). There is also evidence that the plume continued to self-loft after the eruption, reaching 27 km altitude (Khaykin et al., 2022). Figure 6c and 6d show the distribution of aerosols for a 30-day period following the eruption. We restrict the analysis to 30°-60°N where the most of the Reikoke aerosols occurred. In addition to the background distribution (Fig. 6a), there is a band of high extinction observations (> 10$^{-3}$ km$^{-1}$) with CR values ranging from 0.8 to 4.5 locating the volcanic aerosols. The size and concentration distribution retrieved using our algorithm (6d) indeed shows regions with 0.1
µm sizes with a concentration > 50 particles/cm$^3$.

Figures 7 and 8 show the time series of zonal median retrieved particle size and number density following the Reikoke eruption. The median tropopause height is also plotted, and extinction/size/number density may be contaminated by summer convective systems near and below the tropopause. The eruption cloud is initially at 50° N and moves southward so the aerosols are
detected at more southerly latitudes at a later time. At 50°-60°N (Fig. 7) the aerosol size begins to increase after day 20 between 10-15 km. The number density (Fig. 8) also increases rapidly below 15 km. Between 30° and 50°N the upper boundary of the enhanced particle domain appears to increase in altitude consistent with self-lofting although southward and upward transport by the residual circulation may also have contributed to the altitude increase. At all latitudes after day 80, larger particles move downward as expected with aerosol settling. The smaller size particles begin to settle later in the time series.
Initially, the volcanic signal is less discernible in the number density but becomes more evident after day 80.

The overall plume morphology agrees with SAGE III/ISS measurements shown in Schoeberl et al. (2021). Compared to the background aerosols, our retrieval algorithm may have a larger uncertainty when it comes to volcanic aerosols. The accuracy of the retrieved extinction is linked to the phase function, which is computed assuming a constant particle size. When
calculating the color ratio, any inconsistencies between the assumed and actual particle size are not corrected so this may appear as an anomaly in the following retrievals. Nonetheless, the derived large-size particle of this eruption is consistent with Thomason et al., (2021) and Knepp et al. (2022). A recent study by Wells et al. (2022) used model simulation and OMPS-LP measurements to show that including ash in the model, simulation does provide better agreement with the measurements.





Finally, we note the Fig. 8 a, b region with very large number density above 20 km which appears after day 40. This anomaly is due to an overestimate of the CR. SAGE III/ISS CR values are smaller in this region (see supplementary material) and we suspect that this anomaly is due to issues with the OMPS-LP extinctions at higher latitudes and altitudes (see Taha et al., 2021, Fig. 6).

**4.3 Aerosol changes during Hunga Tonga-Hunga Ha'apai (HT) volcano eruption**

The Hunga Tonga-Hunga Ha'apai (hereafter HT) volcano (20.54 °S, 175.38 °W) erupted on January 15, 2022, and injected volcanic material high into the stratosphere. HT lofted a significant amount of water vapor into the stratosphere. The impact and evolution of lofted water vapor and aerosols on dynamics and radiation can be found in Schoeberl et al. (2022a, b). Surprisingly, the $SO_2$ injection was relatively low considering HT's volcanic explosivity index of 5 (Millán et al., 2022; Carn et al., 2022).


Figures 6e, f show the CR vs AE measurement distribution. The figure's shows data for a month, starting a month after the eruption, and we restrict our analysis to 15°S where most of the plume was trapped. The HT aerosol plume shows a higher frequency of measurements with larger sizes and concentration than the Raikoke plume. The significant amount of water vapor lofted during the HT eruption accelerates the growth of sulphate aerosols (Zhu et al., 2022) so this result is consistent with
their findings.

Figure 9 shows the time series of zonal median retrieved particle size and number density between 12.5°S-17.5°S starting in 2022. The extinction (9a) begins to increase immediately after eruption rising to $5\times10^{-3}$ km$^{-1}$. The median particle size (9b) is basically background size (~ 0.1 μm) between 20 km and 30 km until after the eruption (day 15). The gaseous $SO_2$ rapidly
converts to sulfate aerosol (Zhu et al., 2022) which is expressed as a rapid rise in extinction. The particle size starts to increase from background 0.1 μm to 0.15 μm. Meanwhile, the aerosol number density increases at the same height. The median radius continues growing to over 0.2 μm by mid-February. At that time, the aerosols are present at all longitudes (Legras et al., 2022). The peak in the median radius appears in March at 0.25 μm between 20 km and 26 km. After that, the particle size decreases as the aerosol layer settles. Both Legras et al., (2022) and Schoeberl et al. (2022) argue that the settling rate is characteristic of
0.5 μm radius particles. This size is above the retrieval limit for our algorithm, but the largest size particles are clearly descending from day 120 to day 170.

Figure 10 shows vertical profiles of aerosol particle size distribution (a-d) and number density (e-h) for four different days before and following the eruption. The dates of the four profiles are indicated by the dashed lines in Fig. 9. Prior to the
eruption, the aerosol particle size is less than 0.1 μm below 20 km. Above 20 km, size is roughly ~0.1 μm. Figure 10b shows aerosol sizes 30 days after the eruption. There is a sudden increase in size in the lower stratosphere (19-28 km), and subsequently the size increases with the peak appearing between 22 and 26 km. The number density also increases, but the

largest increase is below 22 km by day 100. The increase in size at ~26 km is consistent with a rapid conversion $SO_2$ to sulfate conversion followed by coagulation, while the increase in number density below 22 km suggests settling. The retrieved median
radius of ~0.2 μm is consistent with (Taha et al., 2022).

## 5 Summary and Conclusion

Satellite remote sensing of aerosols provides a global perspective on their role in climate change due to volcanic and pyroCb injections (Kremser et al., 2016) and as well as basic information on the potential use of aerosols in geo-engineering (NASEM, 2021). The challenge is extracting aerosol size and number density from extinction measurements. In this paper we use the
color ratio (the extinction ratio at two wavelengths) to determine size, and from the extinction we determine the number density. We apply our methodology the OMPS-LP and SAGE III/ISS observations of stratospheric aerosols.

Whereas SAGE III/ISS extinction is a direct measurement of the attenuation of the solar beam, OMPS-LP computes extinction from limb scattered radiation. The OMPS-LP algorithm (Taha et al., 2021) assumes the same size distribution for each
wavelength and determines the extinction by matching the observed radiance with radiance computed using a radiative transfer model. The free parameter in this approach is the number density, thus the OMPS-LP algorithm will retrieve an aerosol number density that varies with wavelength. The robust quantity retrieved is, however, the extinction since it must be consistent with the observed radiance.

Our algorithm follows the approach in Bourassa (2008), Thomason and Vernier (2013), Schoeberl et al. (2021) and Kovilakam et al (2022). Using the SAKATRAN radiative transfer model, we show that the color ratio is independent of the number density and only a function of the particle size for a log-normal function size distribution with a fixed width. Thus, the color ratio can be used to constrain the particle size. With the color ratio estimate of size, we can then use the extinction to determine the number density (Eq. 1). The algorithm cannot distinguish sizes greater than ~0.4 μm because the color ratio approaches
one for particles this large. These particles are usually identified as large aerosols or clouds (Schoeberl et al., 2021). Because we use only two wavelengths, we do not have enough information to constrain bimodal distributions. It may be possible to develop an algorithm that can retrieve a bimodal distribution, but that would require additional extinction measurements.

Overall, our algorithm produces particle size and number densities consistent with *in situ* balloon measurements (Deshler and
Kalnajs, 2022); however, the algorithm tends to pick the larger size if the distribution is bimodal. We also apply our algorithm to SAGE III/ISS measurements and our results consistent with OMPS-LP measurements made at the same time and the same latitude. This result is not surprising since Taha et al. (2021) also showed that extinction measurements generated by the OMPS V2.1 algorithm were consistent with SAGE III/ISS measurements. Both the Taha et al (2021) and our results show that errors in the Mie/Rayleigh scattering angle are generally not significant below about 26 km.






Finally, we apply our algorithm to the recent Hunga-Tonga and Raikoke eruptions showing the evolution of the size and number density for these events. Our measurements are consistent with the reported evolution of the aerosol clouds associated with those eruptions (Wells et al., 2022; Schoeberl et al., 2022; Taha et al., 2022; and Khaykin et al., 2022).


*Code and Data availability.* All the data sets used in this study are from NASA and can be downloaded at no cost. The OMPS-LP L2 data are available at https://snpp-omps.gesdisc.eosdis.nasa.gov/data/ (doi:10.5067/CX2B9NW6FI27). SAGE III/ISS data (https://doi.org/10.5067/ISS/SAGEIII/SOLAR_HDF4_L2-V5.1) data are accessible at the NASA Atmospheric Sciences Data Center. The in-situ balloon data are archived at https://doi.org/10.15786/21534894. Data analysis products shown here
are available from the corresponding author.

*Author contributions.* YW and MS were responsible for the development of the aerosol retrieval algorithm, which is described in this paper. YW was responsible for code improvements and testing. YW and MS wrote the initial draft of the paper. GT participated in the scientific discussion about OMPS-LP data. DZ and AB participated in the scientific discussion about
SASKTRAN simulations. GT, DZ, and AB participated in the scientific discussion in regard to retrieval algorithm. All authors reviewed the manuscript and provided advice on the text and figures.

*Competing interests*. The authors declare that they have no conflict of interest.

*Acknowledgments.* The authors would like to thank the OMPS-LP team for Level-2 data production and the in-situ balloon measurement team led by Terry Deshler at University of Wyoming for providing the data used in this study.

*Financial support.* This research has been supported by the National Aeronautics and Space Administration, Earth Science Division (grant no. 80NSSC21K1965).

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



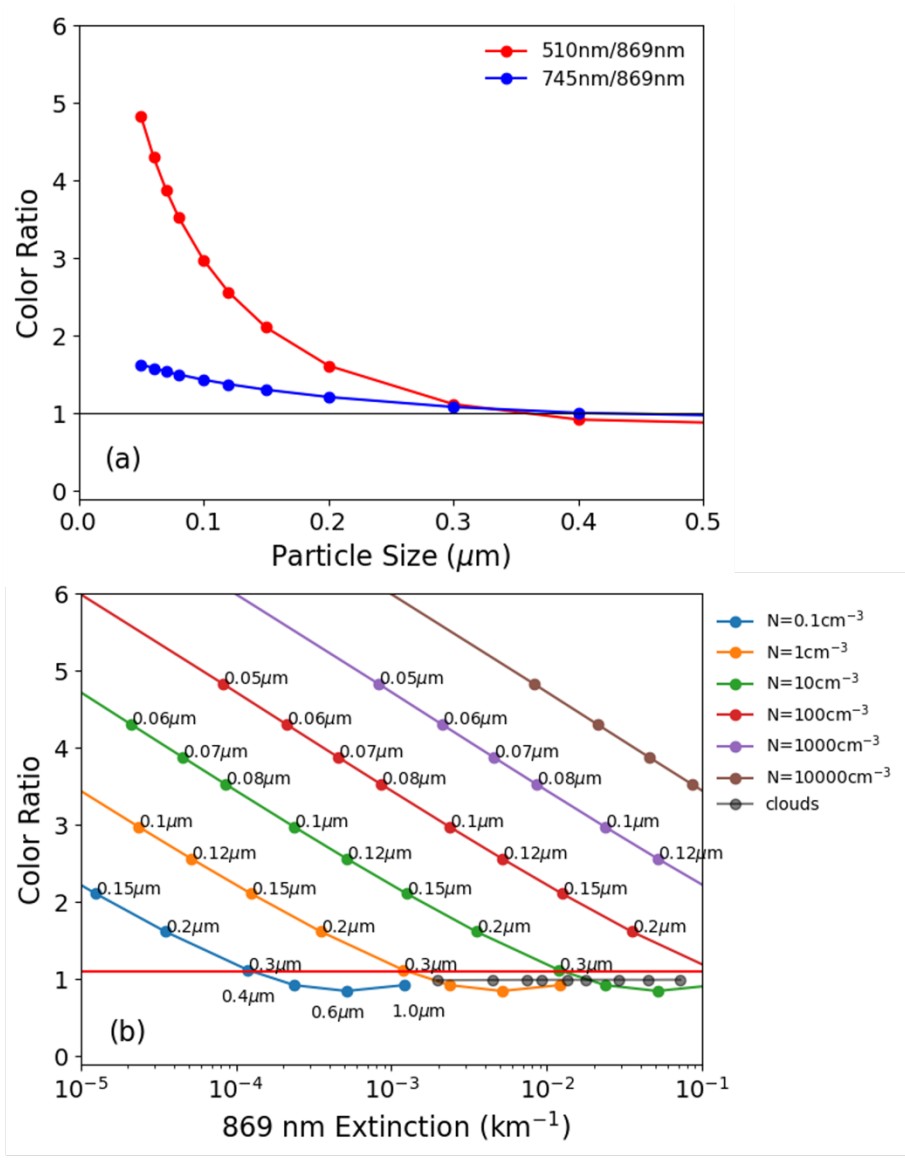


**Figure 1: (a) Aerosol Extinction (AE) Color Ratio (CR) (510 nm/869 nm and 745 nm/869 nm) as a function of aerosol particle size (0.05 μm to 0.5 μm). The black line denotes color ratio of one. (b) The 869 nm AE and 510/869 CR from idealized aerosol (colored lines) and cloud (gray line) scattering experiment with varying number density and particle sizes. The particle radius is shown next to the plotted points. The number density is shown in the legend.**




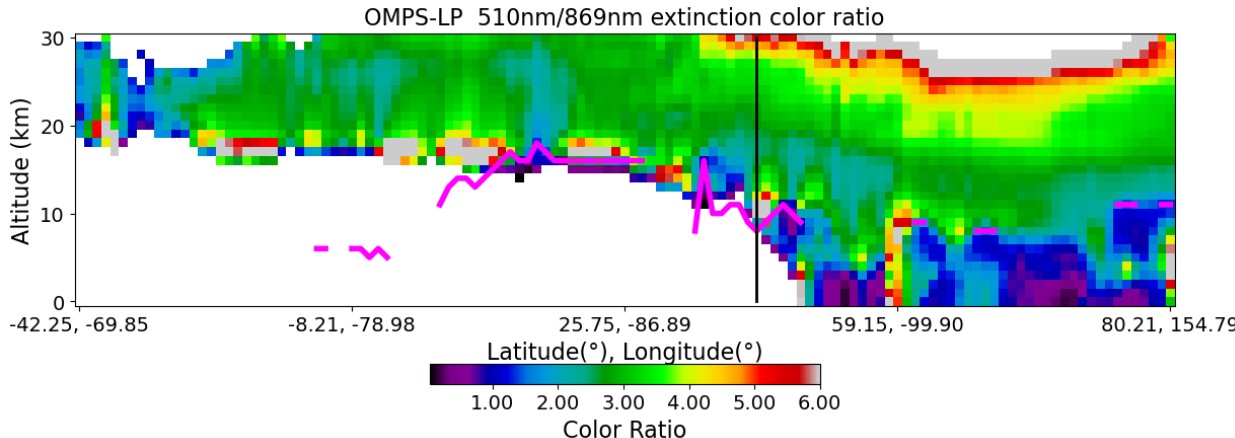


**Figure 2: The OMPS-LP 510/869 extinction color ratio on 9/13/2020. The magenta line shows the OMPS level 2 cloud height. The vertical black line refers to the profile used in Figure 3.**






**Figure 3: The particle size (a) and number density (b) retrieved from a single OMPS profile on 9/13/2020 are shown as red lines. The shaded area is the uncertainty range. For the OMPS number density, we only show the upper limit of uncertainty range as the lower limit extends to zero. The black line is the particle size and number density from the balloon observations on the same day. The left black line is the smaller median size, the right line (when present) is the larger median size from the bimodal distribution.**




**Figure 4: The same as Figure 3, but for the measurement on 8/28/2019.**



**Figure 5: The particle size (upper row) and number density (bottom row) retrieved from OMPS measurements (red color) and SAGE III/ISS measurements (blue color). Band width extends from the first quartile to the third quartile of the data, with a line at the zonal median. For the SAGE data, we use multiple profiles in the latitude range indicated. The OMPS-LP measurements are selected to be near coincident with SAGE profiles. The four columns illustrate the different OMPS-LP scattering angle ranges.**

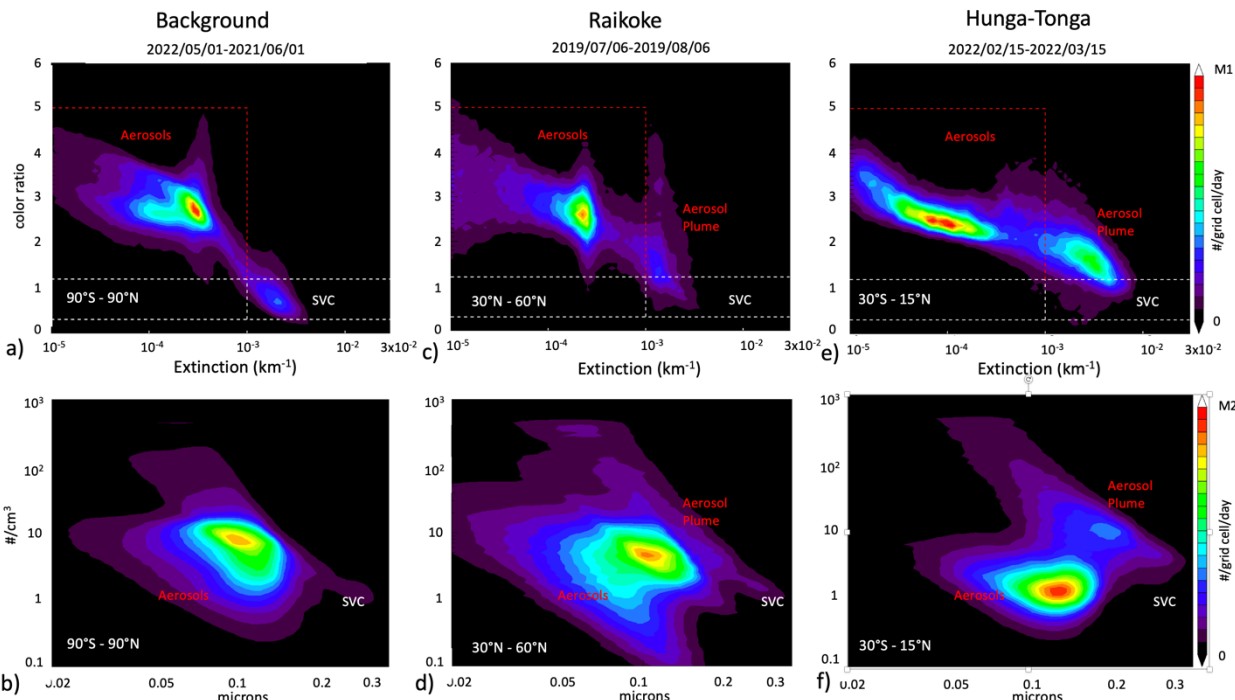


**Figure 6: Upper plots, OMPS-LP extinction vs color ratio (510nm/869nm), lower plots show particle number density vs size using our algorithm. Parts (a, b) non-volcanic background, Parts (c, d) period following Raikoke eruption (e, f) period following the Hunga-Tonga eruption. Dates are shown at the top of the figure; latitude range is shown in each figure. Dashed lines on the top diagrams indicate approximate classification regions, aerosols, aerosol plumes, and subvisible cirrus (SVC). These regions are also identified in the lower figures. The scale range is different for each figure M1=200, 30, 35 & M2=1200, 150, 200 for the color bars, respectively.**


Figure 7: The time series of zonal median particle size around Reikoke volcano eruption. Day 0 is June 21, 2019. Black line is the mean tropopause height.



**Figure 8: The same as Figure 7, but for number density.**




**Figure 9: Hunga Tonga-Hunga Ha'apai time series of zonal median (a) 869nm extinction coefficient, (b) particle size, and (c) number density at 15° S beginning in 2022. The 26 km height is marked as a red line. A vertical line on day 15 indicates Hunga Tonga-Hunga Ha'apai eruption. Four vertical dash lines refer to the four vertical profiles shown in Figures 10 (Day 14, 45, 70, and 100 of 2022).**



**Figure 10: The particle size (upper row) and number density (bottom row) retrieved from OMPS-LP between 12.5 °S and 17.5 °S. The band width extends from the first quartile to the third quartile of the data, with a line at the zonal median. The four columns illustrate the different dates before and after the HT eruption on Jan 15, 2022.**





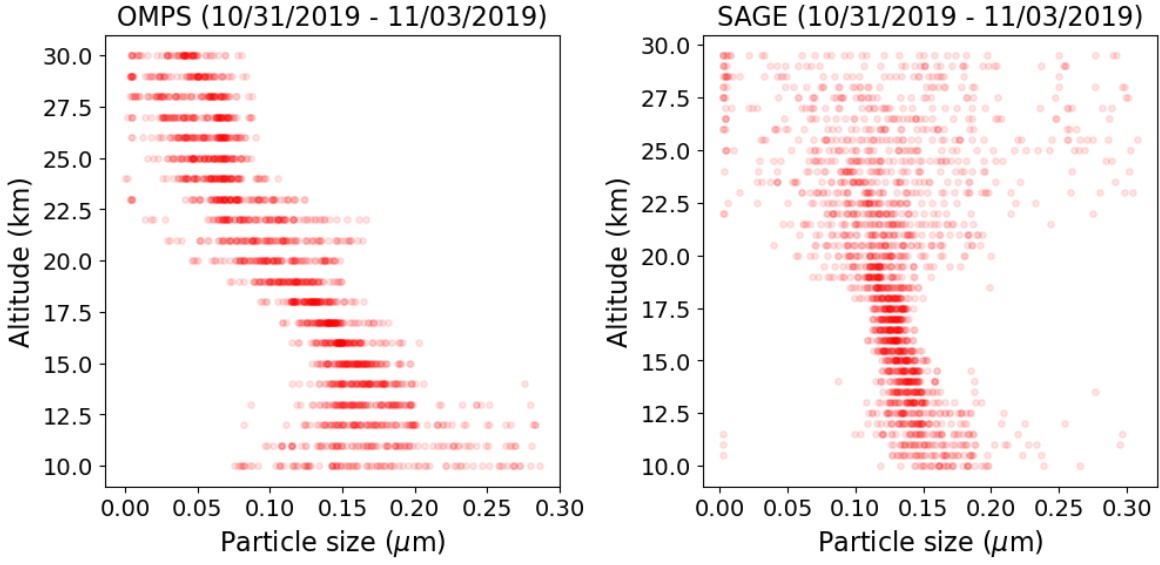


**Figure A1: The retrieved aerosol particle size from OMPS-LP (a) and SAGE (b) between 45 °N and 47 °N during Reikoke volcano eruption (Oct 31, 2019 – Nov 03, 2019).**