# Peer review of "Using OMPS-LP color ratio to extract stratospheric aerosol particle size and concentration with application to volcanic eruptions"

_Atmospheric Measurement Techniques, 2023_

## Referee Comment (RC1)

**Referee report to "Using OMPS-LP color ratio to extract stratospheric aerosol particle size and concentration with application to volcanic eruptions" by Yi Wang al.**

In this manuscript, a simple algorithm suggested for the OSIRIS instrument by Bourassa et al. [1] is applied to OMPS-LP measurements to retrieve the median radius of the log-normal particle size distribution and the number density of the stratospheric aerosols. Authors fix the mode width of the particle size distribution to 1.6 claiming this value to be typical for stratospheric aerosols. To justify this claim authors cite papers of Deshler et al. and Bourassa et al. [3, 2]. In the former paper, the only information on the mode width is given in their Fig. 5 for two single measurements. For the fine mode, it reads 1.26 in the left panel and 1.63 in the right panel of the plot. The second cited paper deals with the merging of ozone data and does not contain any information on the aerosol particle size distribution width. Thus, the claim that the width of 1.6 represents a typical value for the stratospheric aerosols remains absolutely unjustified. Although it is widely known that changes in the distribution width affect the extinction coefficient resulting form the Mie code in a very similar way as changes in the median radius, authors make no attempt in a course of their paper to investigate how a different assumed value of the distribution width would affect their results. The presented validation of the retrieval results is absolutely insufficient for the following reasons: (i) the comparison with both LPC and SAGE III data is made for only two days although much more data is available, (ii) a good agreement with both LPC and SAGE III is seen for the median radius of about 0.1  $\mu$ m while strong deviations are seen for higher values, (iii) no comparison is done for scenes with high volcanic activity, e.g. after Raikoke or Hunga-Tonga eruptions, which are analyzed later in the text. To my opinion the manuscript has to be rejected, as the used scientific method is poorly justified and no reasonable validation of the data has been done. As the quality of data is not properly assessed, it is also unclear if the results illustrating the application of the data to the volcanic eruptions are trustable. I would like to encourage authors to put more efforts in the justification of the validity of their method and validation of the results under different conditions and re-submit the manuscript thereafter.

**Major comments**

- Line 72: "This size distribution is consistent with in situ stratospheric aerosol measurements (Deshler 2003; Bourassa 2014)." as mentioned above, this statement is false. The mode width is consistent with one of the two values reported by [3] and is inconsistent with the other. The second reference is inappropriate here, as it has nothing to do with situ stratospheric aerosol measurements.
- Line 74: "In the discussion below, the 'size' is the median radius of the size distribution." Median radius is absolutely inappropriate quantity for this kind of study because it changes then changing the mode width, even if the extinction coefficients at both wavelengths remain the same.

- Line 74: "Thus, the CR should be nearly independent of the aerosol concentration and only a function of size." this is true for values resulting from the Mie code while the extinction coefficients retrieved from limb-scatter measurements are strongly affected by the scattering and their ratio is not independent of the number density any more. Although the authors claim later in the text that the dependence on the scattering phase function is low, no justification of this statement is provided in the manuscript.
- Line 98: "However, AE is the robust quantity retrieved by the L2 V2.1 algorithm since the AE must match the observed radiance." - again this statement is not true as the radiance is determined not only by the extinction coefficient but also by the scattering phase function. Furthermore, depending on the algorithm, the contribution of the surface reflectance and/or aerosol amount at the reference tangent height might be relevant.
- Line 99: "Thus, if we use the L2 AE at two wavelengths, we have enough information to independently compute a size and number density consistent with the two AE values and independent of the radiative transfer model assumptions about the L2 size distribution." this statement is just wrong. First, authors make an assumption about the mode width. Second, the extinction coefficients at different wavelengths resulting from OMPS-LP L2 retrieval will be different for different model assumptions affecting thus the resulting sizes.
- Line 101: "This approach was also used by Bourassa et al (2008b)." at this point authors should remark that an OSIRIS data set based on this retrieval was never provided by the University of Saskatchewan for a public use. Maybe the coauthors from the University of Saskatchewan might shortly comment why.
- Line 104: "Rieger et al., (2018) compared retrievals between volcanically quiescent periods (small aerosols only) and post eruptions periods characterized by bimodal particle distributions. They did find a retrieval dependence on scattering angles; however the scattering angle error was minor when averaged over similar range of scattering angles." the reference Rieger et al., (2018) is not present in the reference list. It is unclear what authors mean as "the scattering angle error". The dependence of the retrieved aerosol extrinction coefficients on the scattering angle is related to the phase function but this dependence is not relevant in the context of this paper. For the scattering angle dependence the shape of the phase function is relevant while its wavelength dependence is crucial then analyzing the color ratios.
- Line 108: "Since we will compare the retrievals within the same latitudes during the same times, we believe that the biases caused by size-driven Mie phase function error will be small." This conclusion is made w.r.t. the dependency on the shape of the phase function, which is not relevant for this manuscript. It is wrong, however, when applied to the wavelength dependence of the phase function, which is on the contrary highly relevant for this manuscript.

- Sect. 3.1: Why only 2 comparisons are presented? There are certainly more collocations within 10 years operation time of OMPS-LP. Why no comparison is presented for periods of volcanic activity?
- Line 127: "Figs. 3 and 4 show two characteristic profiles." as the results for median radii strongly depend on the assumed mode width of the distribution, an additional comparison for another mode width, e.g. 1.26, must be presented.
- Sect. 3.2: Comparison for just one day cannot be accepted as a validation. There are much more data available. Comparisons for periods of volcanic activity, e.g. after Raikoke and Hunga-Tonga eruptions must be presented. Same as for Sect. 3.1, an additional comparison for another mode width needs to be provided.
- Figs. 3 5: I do not agree with the overall rating of the agreement. In my opinion, a good agreement is seen only if the median radius from SAGE III is around 0.1 while the agreement rapidly degrades if the radius from SAGE III gets larger. Comparisons for scenes with larger particles need to be provided.
- Line 173: "The agreement validates our assertion that errors due to Mie phase function variation with size are minor and that the extinction estimates from the OMPS-LP L2 algorithm are robust." This conclusion seems unjustified to me. As mentioned above, this statement refers to the dependency on the shape of the phase function rather than to its wavelength dependence, although only the latter is relevant for this manuscript. Furthermore, the agreement might be good if the phase function assumed in the retrieval is in a good agreement with the real one and might be worse otherwise. A scenario with larger particles must be considered.
- Conclusions: no word is said about the use of the fixed mode width of the aerosol particle size distribution.

**Minor comments**

- Introduction: Authors do not seem to know anything about European instruments measured aerosol characteristics, e.g. GOMOS, SCIAMACHY.
- Line 62: When talking about the NASA L2 OMPS-LP product, it would be worthwhile to mention that this product uses the Gamma distribution rather than the log-normal one to represent the particle size distribution of the stratospheric aerosols.
- Line 71: "SASKTRAN assumes a log-normal aerosol size distribution with a mode width of 1.6 for spherical sulphate aerosols" this is a bit misleading statement. You assume the mode width of 1.6 not SASKTRAN.

- Line 152: "SAGE will thus report a lower concentration compared to OMPS" this is not necessary true, errors in the retrieved extinction coefficients might also result in wrong median radii. Figs. 3 – 5 provide an impression that a high bias in the median radius is associated with a low bias in the number density and vice versa.
- Fig. 9: Authors should discuss that the figure shows a completely unrealistic behavior of the retrieval below 20 km. The particles are getting smaller and smaller reaching undetectable sizes in panel (d). The pronounced anti-correlation of the median radius and the number density is in accordance with Figs 3 5, 7, 8 a clear indication of retrieval issues.
- Line 267: "The robust quantity retrieved is, however, the extinction since it must be consistent with the observed radiance." this is not completely true for limb-scatter measurements as the retrieved extinction coefficients depend on the assumptions on the aerosol particle size distribution and, depending on the retrieval approach, surface reflectance and aerosol amount at the reference tangent height.
- Line 271: "Using the SAKATRAN radiative transfer model, we show that the color ratio is independent of the number density and only a function of the particle size for a log-normal function size distribution with a fixed width." this statement is misleading, as only the Mie code from SASKTRAN rather than the full radiative transfer modeling was used. A pure usage of the Mie code cannot show anything as this independence results per definition from the used formulas. To show an independence, full radiative transfer modeling followed by the synthetic retrievals needs to be done, which was not the case in the framework of this study.
- Line 284: "errors in the Mie/Rayleigh scattering angle" the statement makes no sense. Most probably you are taking about the shape of the phase function, this is, however, irrelevant in the framework of this study.

**Technical corrections**

• Line 15 and throughout the text: "Reikoke" - should be "Raikoke"

**References**

Bourassa, A. E., Degenstein, D. A., and Llewellyn, E. J.: Retrieval of stratospheric aerosol size information from OSIRIS limb scattered sunlight spectra, Atmos. Chem. Phys., 8, 6375-6380, https://doi.org/10.5194/acp-8-6375-2008, 2008.

Bourassa, A. E., Degenstein, D. A., Randel, W. J., Zawodny, J. M., Kyröl"a, E., McLinden, C. A., Sioris, C. E., and Roth, C. Z.: Trends in stratospheric ozone derived from merged SAGE II and Odin-OSIRIS satellite observations, Atmos. Chem. Phys., 14, 6983-6994, https://doi.org/10.5194/acp-14-6983-2014, 2014.

Deshler, T., Hervig, M.E., Hofmann, D.J., Rosen, J.M. and Liley, J.B.: Thirty years of in situ stratospheric aerosol size distribution measurements from Laramie, Wyoming (41 N), using balloon-borne instruments. J. Geophys. Res. Atmos., 108(D5), 2003

---

## Referee Comment (RC3)

Review of "Using OMPS-LP color ratio to extract stratospheric aerosol particle size and concentration with application to volcanic eruptions", manuscript prepared for AMT by Wang and co-authors.

This manuscript describes an adapted application of an existing method to derive aerosol particle size distribution parameters from measurements of aerosol extinction from the OMPS-LP limb-profiling sensor (e.g. Taha et al., 2021).

The method is applied to analyse vertical and meridional variations in microphysical aerosol properties in background stratosphere conditions and from two volcanic case studies, the 2019 Raikoke and 2022 Hunga-Tonga large-magnitude explosive eruptions,

For the Raikoke case, the derived number concentrations are evaluated comparing to in-situ optical particle counter measurements from high-altitude balloon soundings from North America. The OMPS-derived particle size and number concentrations are compared also to similar colour-ratio analysis from the SAGE-III sensor. The Figures 9 and 10 represent an important and valuable first analysis of the size variations evident across the vertical profile of the Hunga-Tonga stratospheric aerosol enhancement.

The paper is certainly appropriate for AMT, and the analysis of the Hunga-Tonga will be of particular interest with the unexpectedly strong aerosol optical depth. The microphysical parameters are relevant the two hypotheses for the effect -- that the aerosol scattering is amplified by the co-emitted water vapour, and whether there was additional primary emitted aerosol (from the vaporised seawater and/or in-plume-oxidised sulphate).

I have seen that three other revieweers have submitted reviews on the manuscript already, and although I have not referred to their comments when carrying out this review, my comments here are focused on improving the Introduction and Methods section, and in relation to the current interpretation of Figure 3.

Given that this is a manuscript submitted to the specialist journal Atmospheric Measurement Techniques, the methods section requires substantial improvement, and whilst I understand most interest will of course be towards the main scientific results re: the particle size variations, the section 2 requires better summary explanation of the methods.

In some places the scientific writing style needs to be improved for an article in a peer-reviewed journal, avoiding "our algorithm" etc., written in a more formal/impersonal tense. A particular substantial revision, is re: equation 1 of the paper, which as currently written does not convey the approximate nature of the relationship assumed when inferring the large-scale variations in particle number and size.

That said, the article is certainly publishable in Atmospheric Measurement Techiques once the text has been sufficiently improved. The variations shown for the Hunga-Tonga and Raikoke aerosol clouds will be of substantial interest to the stratospheric aerosol community.

Please see below a list of specific revisions to improve the text, and please also check for where there may still be other parts of the text where the wording could be improved.

General Comments

GC1) Finding re: number density independence (text interpreting Figure 1, lines 88-89)

The first of the stated findings in the Abstract (3rd sentence), reached from interpreting Figures 1a and 1b, is not sufficiently demonstrated. The text on lines 88-89 states "This figure shows that the particle size is only a function of CR, and is independent of the number density." There are 2 stated findings in the sentence about particle size, and both are questionable, unless clarified to a specific context.

The upper Figure (1a) shows the curve in colour ratio with particle size from Mie calculations, essentially presenting how much larger the aerosol extinction is at the shorter of the two wavelengths, compared to the longer wavelength, comparing 510nm & 745nm aerosol extinction to that at the reference wavelength of 869nm.

The lower Figure (1b) shows how a set of assumed number concentrations translate into aerosol extinctions at 510nm and 869nm wavelengths, for a range of assumed median sizes.

The reasoning for why the Figure shows this shows one can conclude the number concentration is independent of the number density is far from clear.

The methodology in the paper assumes this to be the case, within a particular range of particle sizes (e.g. particle sizes sufficiently scattering at the corresponding wavelength [e.g. above some threshold value in extinction-cross-section at that wavelength]).

But the text is not correct to state that can be inferred from what is shown in the Figure.

I suggest to delete that text on lines 88-89, and re-write the 3rd sentence of the Abstract that states this to be a finding of the study (lines 10-11).

GC2) Statements re: methods too general or unclarified

The sentence from GC1) is an example of several statements within the manuscript (including within the Abstract) where results are stated too generally, with insufficient communication of the specifics.

Given that this manuscript is within a specialist journal such as Atmospheric Measurement Techniques, the scientific writing on the methods needs to be quite precise.

Whilst I understand that the text describing equation 1 is aiming to present the basis of the Bourassa et al. (2007,2008) method, the explanation on lines 75-83 need to be improved.

For example the sentence on line 76 states "In computing the color ratio of aerosol extinction, the number density cancels out". Whilst that could be OK within a paragraph describing a methological description, here this appears more prominently, and out of that context. My suggestion here is simply to delete this, expression, since it is part of the methodology already described comprehensively in Bourassa et al. (2007,2008).

See specific revisions SR9 and SR10

GC3) Comparisons to balloon-borne laser particle counter measurements (Section 3.1)

This is the other part of the text where the method needs to be better explained (given this is submitted for an Atmospheric Measurement Techniques paper)

The text on lines 125-126 need to provide the location of the sounding compared to, and the specific size-cut for the particle number shown in the black line in Figure 3b. (this information to be re-stated also in the Figure caption).

The terminology can be confusing because the Wyoming laser-OPC (WL-OPC) was developed at Boulder (see Ward et al. 2014) and the new lightweight OPC system is called L-OPC (see Kalnajs and Deshler, 2022).

The cavity-laser OPC is described in Ward et al. (2014), with multiple size channels, down to 75nm radius (75, 150, 250, 500nm, and 1.0, 2.5, 5.0, and 15.0 microns).

For these comparisons to the OMPS-LP aerosol extinction, I am assuming the 75nm radius channel is shown, but this is important considering also that the original OPC40 and OPC25 only measured to 150nm particle radius (see Deshler et al., 2019).

Please add, within the text on lines 125-126, and the caption to Figure 3, the minimum particle size for the size-resolved number concentration shown.

Given the Mie scattering curves will of course vary for the different wavelengths considered, the minimum size is an important issue here.

Related to this a suggestion is to add a dashed line for the R>150nm number concentration (and possibly also the 250nm line, in dot-dashed or so).

The overestimation shown in the 16-18km altitude-range could potentially be due to only some proportion of those R>75nm particles being measured, even at the shorter of the two OMPS-LP wavelengths. I appreciate this is a retrieval, but then the issue of what particle sizes are represented within the two aerosol extinction metrics within the color-ratio particle size method probably justifies considering an uncertainty-range across more than 1 of the size channels (particle size cuts) in the comparisons.

List of specific revisions

SR1) Abstract, lines 8-9 -- This initial text "We have developed an algorithm" might make some readers assume the MS is to explain some development of a new algorithm, but the MS is rather applying an existing method already developed for the OSIRIS data (Bourassa et al., 2007, 2008), to OMPS measurements, comparing to in-situ measurements for two recent volcanic case studies. For this initial sentence better to put the object of the sentence (derive the size...) at the start of the sentence, with also referring more generally to "aerosol microphysical parameters" rather than the specifics in this initial sentence.

Please change "We develop an algorithm that uses the aerosol extinction at two wavelengths..." instead to "We apply an existing method to derive aerosol microphysical parameters from

OMPS dual-wavelength aerosol exinction measurements, to analyse particle size variations for two recent stratospheric volcanic case studies." or similar.

SR2) Abstract, lines 9-10

With the above re-wording, the first part of the 2nd sentence of the Abstract is already integrated into the above re-worded 1st sentence, and the latter part re: SAGE should be stated later, being clear this is SAGE-III on ISS.

SR3) Abstract, line 10 -- This 3rd sentence of the Abstract also needs to be re-worded, with the current text "We show that the color ratio between two wavelengths is insensitive to number density." suggesting the manuscript is presenting this as a finding from the study.

Figure 1 is very useful in showing the variation with particle size and relative magnitudes of monochromatic aerosol extinction at 3 specified wavelengths, for a log-normally distributed liquid aqueous sulphuric acid solution aerosol population (for an assumed refractive index spectrum and water content/weight-percent).

However, as is explained in General Comment 1 above, the text on lines 88-89 is not correct in stating the Figure shows the color ratio is independent of number density. I understand this is an assumption within the methods explained in the Bourassa et al. (2007, 2008) studies, and for a particular range of particle sizes, it's a reasonable assumption to make. However the Abstract should not present as a finding from the study.

Also, the "and thus" in the 2nd part of the sentence (with the current wording) does not follow, and suggests a misunderstanding re: what is assumed in the method, with the wording "We show the color ratio between two wavelengths (e.g. 510nm/869nm) is insensitive to aerosol concentration, and thus can be used to derive aerosol size assuming a log-normal size distribution."

Within the re-constructed sentence, the color ratio metric also needs to better communicated, a suggested re-wording to have "of aerosol extinctions at" in place of "between", and change "(e.g. 510nm/869nm)" instead to "(510mm and 869nm)".

My specific suggestion for the 2nd part of this sentence is to delete "and thus can be used to", and also change "We show that" instead to a re-worded sentence similar to below:

"The color ratio between aerosol extinction at two wavelengths is used to derive from satellite measurements large-scale particle size variations within volcanic aerosol clouds in the stratosphere."

SR4) Abstract, lines 12 -- Re-word "With the size and extinction, we can compute a number density consistent with both wavelengths". The particle size is being inferred from the two-wavelength aerosol extinctions, not measured directly. With the re-constructed preceding sentences (from SR3), suggest to instead have this sentence be to explain some greater specifics in the method, prior to its application for profiling the size distribution of the Raikoke and Hunga Tonga aerosol clouds.

A suggested re-wording "With the size and extinction, we can compute..." instead with "As a further microphysical Consistent with the two extinction

SR5) Abstract, lines 13-14 -- Please re-word this sentence to be more specific re: the comparisons between the particle size variations from OMPS-LP and those derived from SAGE-III on ISS, giving some specifics about how well the two size products compare.

SR6) Introduction, lines 21-22 -- change "have been connected to short-term changes in climate" instead to "major eruptions causing substantial short-term changes in climate", or similar.

The first sentence (of this 1st paragraph of the introduction) mentions the volcanic impacts on climate, but this adaptation of the sentence is then consistent with the strong (but temporary) radiative forcings after historical very large-magnitude explosive tropical eruptions.

SR7) Introduction, line 25 -- Please re-word this sentence to be clearer what is meant by "dust" in this context ("Volcanic ash, Pyro-CB smoke and dust") -- presumably it is cosmic dust (i.e. meteoric aerosol) that is meant here, right? Suggest to have the re-worded text provide this information in relation to citing a recent observational studies for each of these non-sulphate stratospheric aerosol constituents. Suggestions are Vernier et al. (2016), for volcanic ash, Khaykin et al. (2020) for pyro-Cb smoke, and Schneider et al. (2021) for meteoric aerosol.

With this re-wording the text on line 26, suggest to cite paper also for the volcanic ash heating (Muser et al., 2020), and cite the Yu et al. (2019) for the wildfire smoke heating effect. The cosmic dust (meteoric smoke) may be present at sizes smaller than for wildfire smoke and volcanic ash, also some components dissolving into the sulphuric acid solution aerosol

(see James et al., 2023), the heating effect is primarily associated with the smoke and ash.

SR8) Introduction, lines 32-34

particles

Re-word "has provided global monitoring of the stratospheric aerosol layer since 1975" because the SAGE and SAM-II record only began in 1979 (see McCormick et al., 1977). The original testing of the SAM sensor on the Apollo Soyuz mission was in 1975, but the global monitoring only began in 1979.

Re: the text on the solar occultation methods, the so-called "onion-peeling" retrieval method was originally developed in the 1960s (Edward Ney group at the University of Minnesota), and it will help retain the heritage of the instrument development to cite the Pepin (1969) report that first documented the technique, and the Rosen et al. (1969) which shows (Figure 4) the only peer-reviewed paper showing the initial application of the methods to balloon-borne solar extinction measurements, and later further developed for application to the first satellite

measurements of the stratospheric aerosol layer (the SAM, SAM-II, McCormick et al., 1979), and then SAGE, SAGE-II and SAGE-III series of satellite measurements.

SR9) Section 2, line 76-78

Add ", at wavelength \$\lamda\$, " after "The aerosol extinction" and add subscript lamda symbol to the two wavelength-dependent symbols in equation 1 (sigma and AR).

In addition, please change the abbreviation "AE" for aerosol extinction, as this may initially confuse some readers, given the related size-associated aerosol metric "Angstrom exponent". Please change all occurrences of "AE" instead to either "k" (used in the Bourassa et al. (2007, 2008) papers) or other terminology (the letter b is used, subscript "ext" to denote aerosol extinction in Seinfeld & Pandis (2006).

It also needs to be stated explicitly that these calculations are based on an assumed log-normal size distirbution, with also the associated water content (sulphuric acid and water composition).

As explained in General Comment GC2, please delete the two short sentences lines 77-78 as this statement does not follow from what is shown. The information and associated assumptions are explained within context in the Bourassa et al. (2007,2008) papers.

SR10) Section 2, lines 82-83

As mentioned in General Comment GC2, the text "...but as the two wavelengths approach each other" is too colloquial and needs to be formalised. Also, the "Any two distinct wavelengths can be chosen" at the start of the sentence also does not need to be stated. Suggested restructuring to better explain the issue.

Suggest to re-word the original text:

"Any two distinct wavelengths can be chosen for CR, but as the two wavelengths approach each other Fig. 1a shows that the CR gradient decreases and thus the uncertainty in the retrieved size increases."

instead to

"For OMPS-LP wavelengths closer to the reference 869nm wavelength, the gradient in the colour ratio curve is less steep, causing an increased uncertainty in the retrieved particle size."

It is worth noting also that the 869nm aerosol extinction measurement is "cleanest" retrieval in relation to effects from other species, and this must also be considered in relation to how the errors/uncertainty differ between different color-ratio wavelength-pairs.

SR11) Figure 1 caption, lines 431-435.

Although it is not stated explicitly, my understanding is that these calculations are based on a log-normally distributed aerosol, with geometric standard deviation of 1.6, following the specification within SASKTRAN. This should be stated in the caption to Figure 1.

SR12) Section 2, line 368 -- formatting error re: Loughman et al. (2015) reference.

References

Bourassa et al. (2007) Stratospheric aerosol retrieval with OSIRIS limb scatter measurements, J. Geophys. Res., 112, D10 217, https://doi.org/10.1029/2006JD008079.

Bourassa et al. (2008) Retrieval of stratospheric aerosol size information from OSIRIS limb scattered sunlight spectra, Atmos. Chem. Phys., 8, 6375,Äì6380, https://doi.org/10.5194/acp-8-6735-2008.

James et al. (2023) The importance of acid-processed meteoric smoke relative to meteoric fragments for crystal nucleation in polar stratospheric clouds Atmos. Chem. Phys., 23, 2215,Äi2233, https://doi.org/10.5194/acp-23-2215-2023

Khaykin et al. (2021) "The 2019/20 Australian wildfires generated a persistent smoke-charged vortex rising up to 35 km altitude" Comms. Earth Env., https://doi.org/10.1038/s43247-020-00022-5

McCormick et al. (1978) Satellite studies of the stratospheric aerosol Bulletin of Amer. Meteorol. Soc., vol. 60, no. 9, 1038-1046. https://doi.org/10.1175/1520-0477(1979)060<1038:SSOTSA>2.0.CO;2

Muser et al. (2020) Particle aging and aerosol,Äiradiation interaction affect volcanic plume dispersion: evidence from the Raikoke 2019 eruption, Atmos. Chem. Phys., 20, 15015,Äi15036, https://doi.org/10.5194/acp-20-15015-2020

Pepin T. J. (1969) "The Use of Extinction from High Altitude Balloons as a Probe of the Atmospheric Aerosols", Univ. Minn. progress report, Oct, 1969 https://apps.dtic.mil/sti/pdfs/AD0696527.pdf

Pepin, T. J. (1977) "Inversion of solar extinction data from the Apollo-Soyez test project Stratospheric aerosol measurement (ASTP/SAM) experiment", volume 1, 529-544, NASA report SP-442. https://ntrs.nasa.gov/citations/19780009145

Rosen, J. M. (1969) "Stratospheric dust and its relationship to the meteoric influx", Space Science Reviews, vol. 9, 58-89, https://doi.org/10.1007/BF00187579 (see Figure 4).

Schneider et al. (2021) " Aircraft-based observation of meteoric material in lower-stratospheric aerosol particles between 15N and 68N", Atmos. Chem. Phys., vol. 21 989-1013, https://doi.org/10.5194/acp-21-989-2021

Seinfeld and Pandis (2006) Atmospheric Chemistry and Physics: From Air Pollution to Climate Change, 2nd Edition, Wiley publisher. John H. Seinfeld, Spyros N. Pandis

Vernier et al. (2016) "In situ and space-based observations of the Kelud volcanic plume: The persistence of ash in the lower stratosphere", J. Geophys. Res., vol. 121, 11,104,Äìll,118, http://doi.org/10.1002/2016JD025344.

---

## Community Comment (CC2)

The authors present a methodology for estimating aerosol radii (mode radius of the lognormal distribution) using a single wavelength combination (510 nm and 869 nm) from the OMPS-LP record. The authors introduce this method and provide a comparison with balloon-borne optical particle counter data as well as a coincident SAGE III/ISS profile as validation of the technique. Having demonstrated the validity of this technique the authors then apply it to data collected after major disruptions (i.e., the 2019 Raikoke eruption and the 2022 Hunga Tonga eruption).

**Major Concerns with this Manuscript**

I have two major concerns about this publication:

1. The general methodology is not new. A very similar method was published in 1983 by Glenn Yue (DOI: 10.1364/AO.22.001639) and another method (using 2 ratios) was published in 2021 by Felix Wrana (DOI: 10.5194/amt-14-2345-2021)). Both of these papers (and numerous others in between) deal with extinction ratios from SAGE instruments, but the gist is the same.

2. The methodology itself is fundamentally flawed and the derived products are wholly unreliable. I present a simple model below to demonstrate this unreliability. The authors assumed that the information content of 1 extinction ratio is sufficient to derive a valid estimate of particle size, but this only holds true if the distribution width is fixed and the measurement error is sufficiently small; both are invalid assumptions. While an assumed distribution width of 1.6 is a good estimate, fixing the width to that value (or any other value) imposes an artificial constraint on the solution space and inevitably biases the inferred radii and number density results. Ultimately we have to recognize that we know very little about the atmosphere (the width could be 1.2, or it could be 1.9; both are very realistic) and forcing the distribution width to 1 specific value is wrong.

Given the flawed methodology and the lack of novelty it is difficult to see how this paper makes a substantive contribution to the scientific literature (we can get the same information from a paper written 40 years ago). However, I recognize that I may have missed some nuances of their method so I ask for clarification on several points.

**Specific Comments**

1. A comment on the references: The authors cited many manuscripts that do not correspond to the text they supposedly support. For example, on lines 54–56 the authors state that their method of determining particle size is based on 4 previous publications and all of the cited papers deal with cloud identification and filtering, not determination of size. Further, the Bourassa et al. 2007 paper does not seem to fit at all. The same is true for the Bourassa 2014 paper cited on line 73. Bourassa 2014 has to deal with stratospheric ozone trends. Perhaps the authors intended to cite Bourassa 2008 instead, but even that paper does not support their text (Bourassa 2008 cites Deshler et al. 2003, but the context within which Bourassa 2008 is cited here indicates that they actually did in situ measurements, which they did not).

2. Line 10: The authors claim that they demonstrate that extinction ratio is insensitive to aerosol concentration. This is nothing new and can be observed by looking at the corresponding equations.

3. Line 51: As stated in this paper the OMPS-LP retrieval assumes a size distribution to obtain the extinction products. The authors then used the extinction products to infer a size distribution, which makes a cyclical process. What if the assumed size distribution used in the OMPS-LP processing was different, would that change the derived size? What is the level of correlation between the assumed size distribution and the derived particle size?

4. Line 60 (all of section 2): It is unclear whether the authors accounted for the uncertainty in the OMPS-LP products. Given the content of some of their figures I assume they did, but it is never explicitly stated (see comment below regarding error propagation).

5. Line 72: It is unclear why the authors assumed a distribution width of 1.6. Granted, this value makes for a reasonable first guess, it used in SASKTRAN, and was used by Bourassa et al 2008 (the authors cited Bourassa 2014). However, Deshler et al. 2003 in no way claims that 1.6 is the only value that should be used. The Deshler et al. 2003 paper presents a figure (Fig. 5 panel B) that contains a derived size distribution from 1 altitude (20 km) of 1 profile; this in no way supports the use of a static distribution width of 1.6. This is a key point.

   (a) The authors took this value of 1.6 (collected during the "background period" at 20 km), failed to account for the natural variability of this value and made the assumption that it never changes. This is particularly a problem when the authors use the same distribution width after major eruptions.

   (b) The distribution width in the atmosphere is not static. It changes with season, altitude, latitude. The width is highly variable even when the atmosphere is not substantially impacted by volcanic and/or pyroCb activity (see Fig. 1). While the assumed width of 1.6 may be reasonable, it cannot be assumed to be static.

[Figure]

Figure 1: Profile of mean ($\pm$ 1-sigma) distribution widths from the University of Wyoming OPC record. Data collected during the quiescent period (2000–2010).

(c) The University of Wyoming dataset reports an uncertainty in distribution width of
±20%. Therefore, even if Deshler et al. 2003 said that the width is consistently 1.6
that would still leave a range between 1.56 and 1.63. If we make some assumptions we
can model the expected behavior: assume sigma error is fixed at 20% (per Deshler et
al. 2003) and the measurement error (propagated uncertainty in the extinction ratio)
is *only* 5% (this is conservative as Taha et al. 2021 report accuracy/precision on the
order of ±20%). Here (Fig. 2) we see the range of mode radii that produce extinction
ratios that fall within these uncertainty bounds (everything from ≈40 nm through 190
nm). The question then is "Which mode radius is the 'real' one, or which do you pick?"
Each radius is a viable solution so the uncertainty in the authors' estimate is far larger
than they show.

If the authors were to use a realistic uncertainty in their estimate of distribution width
(e.g., along the lines of the atmospheric variability shown in Fig. 1) and were to account
for the propagated measurement uncertainty then they would see the solution space ex-
pand quite rapidly. This point cannot be overstated: the distribution width is highly
important and is far from static, fixing the width to 1.6 (or any other value) imposes an
unjustified constraint on the solution space and introduces bias in the inferred radius
estimates as well as the corresponding number density estimate. It is for this reason
that I see the method as fundamentally flawed.

The model I present in Fig. 2 is overly conservative and presents a best-case sce-
nario. The point I'm getting to is: Even under these best-case scenarios we cannot
make a definitive statement about the particle size. The requisite information content
is not there.

[Figure]

Figure 2: Distribution of radius solutions when the error in distribution width is fixed at ±20%
and the uncertainty in extinction ratio is fixed at ±5%.

6. Line 78: The authors suggest that the "CR" is "only a function of size" and I am uncertain
of what is meant by that. The CR is a function of particle size distribution parameters (both
mode radius and distribution width).

7. Line 79: Could the authors please explain why the 510/869 combination was chosen instead of 510/997? The 510/997 combination would expand the "usable" range from 0.4 $\mu$m to $\approx$0.5$\mu$m.

8. Lines 80–81: "This CR – size relationship allows us to infer the median aerosol particle radius up to $\approx$0.4 $\mu$m." This will vary depending on the distribution width.

9. Lines 99–100: "Thus, if we use the L2 AE at two wavelengths, we have enough information to independently compute a size and number density..." This is not true as demonstrated above. Even with multiple extinction ratios you would not have enough information to definitively determine particle size. The best we can do is report a range of radii.

10. Lines 103–104: In the previous paragraph the authors stated that their method was "independent of the radiative transfer model assumptions", but now they state this is a potential source of error. Could the authors please clarify?

11. Lines 129–130: The authors stated "The uncertainty ranges of OMPS-LP retrievals are calculated from the extinction coefficients (AE), using the formula below". The context of this paragraph led me to believe that Eq. 2 was used to calculate the error in derived mode radius...but this is just an error propagation. Could you authors please clarify how this equation was used to generate the errors in their Fig. 3 & 4?

12. Section 3.2: The purpose of this section is unclear. I see 2 possibilities:

    (a) Do the authors present this as corroboration of their size estimate? If so, then this fails as all this demonstrates is that the SAGE III/ISS extinction ratios are in agreement with those of OMPS-LP.

    (b) Do the authors present this as validating the OMP-LP extinction ratios (i.e., since the derived radii agree with the radii derived from SAGE III/ISS then OMPS-LP and SAGE III/ISS must be reporting the same extinction ratios)? I wonder because later in this section the authors state "The agreement validates our assertion that errors due to Mie phase function variation with size are minor and that the extinction estimates from the OMPS- LP L2 algorithm are robust." (lines 173–175). If this was their intent, then why is this needed and why does this fall within this paper (it seems a stark departure from the stated intent)? Also, didn't Taha et al. 2021 already do this validation?

I thank the authors for taking the time to read my comments and look forward to their feedback.

---

## Community Comment (CC4)

Dr. Wang, thank you for the early and thorough response!

My overall impression is that the initial response provided by the authors does not resolve the original problems with the methodology and I struggle to see the validity and utility of this method. Fundamental problems that render the inferred particle sizes unusable remain. These issues deal, to a great degree, in how the authors interpret and use a single figure from Rieger et al. 2018. The authors take that figure as definitive and prescriptive (a cursory look at Rieger's Fig. 6 leaves no ambiguity that assuming a distribution width of 1.6 with an error of  $\pm 0.2$  throughout the atmosphere is categorically wrong), fail to recognize the difference between the fine and coarse modes, and fail to address the real variability of particle size distribution parameters as measured by the University of Wyoming's optical particle counters (UWY OPC; this was the dataset Rieger et al. used to create their Fig. 6). If the authors were to expand the uncertainty in the distribution width beyond 1- $\sigma$ , which only accounts for 68% of the atmospheric variability, then the uncertainty in the inferred particles becomes even more egregious. Finally, even with their overly conservative error estimates the particle size profile in Rieger et al.'s Fig. 6 is still better than what the authors predict; if the authors take Rieger's distribution width as canonical then why not use Rieger's radius profile instead of inferring it from OMPS. I think, in many cases, using Rieger's radius profile would yield a narrower solution space. Overall, it is difficult to see the utility of this method and its shortcomings are not accurately brought forward to inform the reader of its limitations. Despite the numerous fundamental flaws, as described below, the authors fail to communicate any of these issues or limitations to the reader.

The authors' response to my comments are in black, my additional comments are in blue. If the authors resubmit would they kindly address both of my communications in their revised manuscript?

**Major Concerns with this Manuscript**

- 1. As stated in my original review this methodology is not new. Variations go back to at least 1982. The authors agreed with this but suggested that the novelty lay in the application to OMPS data collected in the aftermath of 2 recent eruptions. The concerns I have with this are:
  - (a) Application of an old method to a new dataset is, by itself, not interesting. The official reviewers and editor have the final say in this, but to me it is not interesting.
  - (b) The authors used this method to look at the size evolution after 2 recent eruptions. However, this was done without a thorough evaluation of the accuracy of the method. I note that another reviewer had similar concerns.
  - (c) The authors failed to address the instability of the solution space and account for the natural variability of aerosol size distributions within the atmosphere. The current model is based on the assumption that the distribution width is fixed and can never change. This not only neglects the natural variability but also the uncertainties in the OPC data. I still view this as a fundamental flaw and explain why below.
  - (d) As explained in Taha et al. 2021, OMPS' 510 nm channel should only be used between 20–24 km and only in the northern hemisphere. The authors used this data throughout the profile in both hemispheres. Could the authors please explain to the reader why the recommendation of Taha et al. was not followed?

2. "We concur with your assertion that the distribution width may vary, but numerous in situ measurements have constrained the range of widths and models of the size distribution for ambient condition (Rieger et al., 2018)."

The claim that "numerous in situ measurements have constrained the range of widths and models of the size distribution" is not true. These datasets describe the natural variability, but in no way limit the variability. This is an important point: Fig. 6 of Rieger et al. (plot of average OPC profiles) is *not* prescriptive, rather it is descriptive. Further, the uncertainty in distribution width, as provided by Rieger et al., of  $\pm 0.2$  is just the standard deviation of the OPC's estimated widths *at a single altitude* (20 km): this error becomes *much* larger lower and higher in the atmosphere. Further, this only represents 68% of the real atmospheric variability (at 20 km). The consequence of this is that 32% of the real distribution widths at 20km will be greater than  $\pm 0.2$  (again, this plus/minus value becomes significantly larger the further you get from 20 km). The problem is we do not know which points are closer to 1.6 and which are farther, which makes estimating the mode radius uncertainty challenging. A further complication is the uncertainty in the OPC's estimated errors, which further expands the solution space. In short, the static  $\pm 0.2$  value is not correct, it introduces an unrealistic constraint on the solution space (even at 20 km), introduces systematic bias, and the reader is left with no recognition that any of these issues exist.

- 3. "In the context of this retrieval method, assuming a fixed distribution width is a necessary step and a common approach used in current retrieval algorithms..." This may be common practice, but we are under obligation to evaluate the impact that our assumptions have on our results. The original manuscript ignored this almost entirely and the authors' response to my initial comment failed to fully recognize the impact these assumptions have *and* leave the reader with the impression that these problems do not exist.
- 4. "...as the reviewer notes, 1.6 is a good estimate for the distribution width." That is correct. A value of 1.6 is a good estimate, but that does not mean it is accurate. As stated above, accounting for the natural variability of these widths expands the solution space significantly and this must be accounted for before the reader can have confidence in this product.
- 5. "All remote sensing systems make assumptions about size distributions as noted above." I apologize if this comes across as being "nit picky", but this is not true. My only point here is that if the authors decide to include a comment similar to this in the revised version to please be more nuanced in their meaning.
- 6. "However, Rieger et al. (2018) Fig 6 shows that not all distribution widths are likely, and 1.6 is a reasonable choice."

Rieger et al is not prescriptive and using a static value of 1.6 is demonstrably wrong. For example (looking at Rieger's Fig. 6), 1.6 falls near the mean value at 35 km, but at that altitude the distribution width ranged from  $\approx 1.2-2.25$  (far outside the  $\pm 0.2$ ). At 10 km the mean width ranged from  $\approx 1.6-\approx 2.7$  (it went off the scale so I cannot tell). Finally, at 22.5 km the width ranged from  $\approx 1.1-1.6$ . The point I'm trying to make is that while a value of 1.6 is reasonable for a rough estimate we have to understand that there is a high probability of the width being very different from 1.6 and that the assumed standard deviation (i.e., the  $\pm 0.2$  value) is highly dependent on altitude (especially under perturbed conditions). Further, Rieger's Fig. 6 only shows the mean distribution width  $\pm 1-\sigma$ , which leaves 32% of

the atmospheric variability unaccounted for). In short, the assumed distribution width (1.6) is incorrect, the assumed uncertainty of this width (0.2) is incorrect, and both values were applied statically throughout the profile without regard to the actual atmospheric variability as shown in Rieger et al. 2018 or in the UWY OPC record. This introduces insurmountable errors in the estimated radius, which makes the product unusable.

7. "To summarize, Figs. 1, 2 quantify the expected impact of CR uncertainty and distribution width uncertainty on size using values from Taha et al., (2021) and Rieger et al (2018). For 1.6 distribution width and color ratios between 2 and 4, the maximum size uncertainty is 20%..."

Thank you for these informative figures. They certainly convey a lot of information. However, I still have some concerns about this analysis for the following reasons: 1. as explained above the distribution width is not consistent at 1.6 (this is even shown in Rieger et al.'s Fig. 6); 2. the uncertainty of with distribution width is not consistent throughout the profile (as shown in Rieger et al.); 3. the error used in your distribution width is fixed at  $\pm 0.2$ , which is only  $1-\sigma$  (at 20 km), meaning this leaves  $\approx 32\%$  of the variability unaccounted for (since the standard deviation expands as you move away from 20 km this  $\pm 0.2$  value accounts for less and less of the variability); 3. Taha et al. 2021 recommend only using the 510 nm channel between 20-24 km and only in the northern hemisphere (presumably the uncertainty becomes untenable outside these regimes). Even though the errors used in the calculation that went into Fig. 1 of your response are overly conservative, we still get a spread in radii that exceeds the values presented in Rieger's Fig. 6. For me, it is hard to determine what value has been gained by using this method instead of just using the statistical representation of the OPC data as Rieger did.

8. "Note the Rieger et al. (2018) also provides size distribution widths for coarse mode particles and the fine mode and coarse mode distribution widths are similar as are the mean distributions (1.6)."

This is a very challenging problem. The coarse mode in Rieger et al. is actually the second mode in a bimodal distribution. To reduce this bimodal system to a single mode, and call the second mode's distribution width representative of the overall width, is not correct. Further, Rieger et al. did use the same "representative" value of 1.6 for the second mode, but the 2 profiles (fine mode vs coarse mode) are substantially different. Remember: Rieger et al. is not prescriptive and their  $1.6\pm0.2$  values are *not* valid throughout the profile and cannot be taken as a one-size-fits all approach. Details must be paid to the real variability in the atmosphere and not limited to a previous author's assumptions. Again, I agree that dealing with this second mode is challenging, but the argument put forth by the authors (quoted above) is categorically wrong. The consequence of their assumptions is that not only are the inferred particle radii under background conditions wrong, but they are even worse following major events.

9. "Section 3.2 aims to assess the impact of varying scattering angles on retrievals. This analysis is also related to Reviewer comment 3, and partly evaluated the error source from that. Yes, the comparison essentially compares the color ratio between OMPS-LP and SAGE III/ISS, but the purpose is different. Additionally, it is valuable to include a section comparing the retrieved particle sizes, rather than directing readers to seek out the color ratio comparison from other sources. We will make the goals of this section clearer."

The entire paper, to this point, has been dedicated to inferring particle sizes from OMPS

color ratios. Introducing particle estimates from SAGE data in order to "...assess the impact of varying scattering angles..." provides an unforeseen transition. This will be confusing to the reader. My main concern here is that taking the additional step to convert SAGE data to particle sizes introduces an unnecessary layer of obfuscation that only decreases the validity of this comparison. It would be far more meaningful to do a direct comparison of the SAGE and OMPS extinction products as was done by Taha et al. 2021. In short, it is difficult for me to see how this section benefits the paper and it seems this type of comparison has already been done.

Again, I thank the authors for taking the time to read my comments and look forward to their response.

---

## Author Comment (AC1)

We would like to thank the reviewer for their comments on the paper. Below is our response.

1. The general methodology is not new. A very similar method was published in 1983 by Glenn Yue (DOI: 10.1364/AO.22.001639) and another method (using 2 ratios) was published in 2021 by Felix Wrana (DOI: 10.5194/amt-14-2345-2021)). Both of these papers (and numerous others in between) deal with extinction ratios from SAGE instruments, but the gist is the same.

R: Thank you for bringing these additional papers, in addition to those we cited, to our attention. We apologize for the oversight and will add them to the manuscript. While we acknowledge that the general methodology of using color ratio to retrieve particle size is not new, we believe that applying this method to OMPS-LP and focusing on recent volcanic eruptions represents a valuable contribution to the field.

2. The methodology itself is fundamentally flawed and the derived products are wholly unreliable. I present a simple model below to demonstrate this unreliability. The authors assumed that the information content of 1 extinction ratio is sufficient to derive a valid estimate of particle size, but this only holds true if the distribution width is fixed and the measurement error is sufficiently small; both are invalid assumptions. While an assumed distribution width of 1.6 is a good estimate, fixing the width to that value (or any other value) imposes an artificial constraint on the solution space and inevitably biases the inferred radii and number density results. Ultimately, we have to recognize that we know very little about the atmosphere (the width could be 1.2, or it could be 1.9; both are very realistic) and forcing the distribution width to 1 specific value is wrong.

R: We concur with your assertion that the distribution width may vary, but numerous *in situ* measurements have constrained the range of widths and models of the size distribution for ambient condition (Rieger et al., 2018). However, we disagree that the methodology is fundamentally flawed. In the context of this retrieval method, assuming a fixed distribution width is a necessary step and a common approach used in current retrieval algorithms, and, as the reviewer notes, 1.6 is a good estimate for the distribution width. In response to the longer comment by the reviewer below we have quantified the impact of the distribution width assumptions as described in response to comment 7.

3. A comment on the references: The authors cited many manuscripts that do not correspond to the text they supposedly support. For example, on lines 54–56 the authors state that their method of determining particle size is based on 4 previous publications and all of the cited papers deal with cloud identification and filtering, not determination of size. Further, the Bourassa et al. 2007 paper does not seem to fit at all. The same is true for the Bourassa 2014 paper cited on line 73. Bourassa 2014 has to deal with stratospheric ozone trends. Perhaps the authors intended to cite Bourassa 2008 instead, but even that paper does not support their text (Bourassa 2008 cites Deshler et al. 2003, but the context within which Bourassa 2008 is cited here indicates that they actually did in situ measurements, which they did not).

R: Thanks for pointing this out. We will correct Bourassa et al. (2007) to Bourassa et al. (2008b) and fix other citations.

4. Line 10: The authors claim that they demonstrate that extinction ratio is insensitive to aerosol concentration. This is nothing new and can be observed by looking at the corresponding equations.

R: We do not intend to claim this is a new finding. We will rephrase the sentence to make it clearer.

5. Line 51: As stated in this paper the OMPS-LP retrieval assumes a size distribution to obtain the extinction products. The authors then used the extinction products to infer a size distribution, which makes a cyclical process. What if the assumed size distribution used in the OMPS-LP processing was different, would that change the derived size? What is the level of correlation between the assumed size distribution and the derived particle size?

R: See information above. The OMPS-LP retrieval assumes a size distribution to compute an extinction consistent with the observed radiance. To achieve this result, the algorithm varies the concentration. Thus, the aerosol concentration varies with wavelength, which is unphysical. The fundamental retrieved quantity is the extinction. Using the two wavelength extinction ratio (color ratio), we recompute a consistent size and concentration, the only assumption is the distribution width.

6. Line 60 (all of section 2): It is unclear whether the authors accounted for the uncertainty in the OMPS-LP products. Given the content of some of their figures I assume they did, but it is never explicitly stated (see comment below regarding error propagation).

R: We will add an uncertainty range of retrieval for different distribution width values instead of just giving an error propagation.

7. Line 72: It is unclear why the authors assumed a distribution width of 1.6. Granted, this value makes for a reasonable first guess, it used in SASKTRAN, and was used by Bourassa et al 2008 (the authors cited Bourassa 2014). However, Deshler et al. 2003 in no way claims that 1.6 is the only value that should be used. The Deshler et al. 2003 paper presents a figure (Fig. 5 panel B) that contains a derived size distribution from 1 altitude (20 km) of 1 profile; this in no way supports the use of a static distribution width of 1.6. This is a key point.

(a) The authors took this value of 1.6 (collected during the "background period" at 20 km), failed to account for the natural variability of this value and made the assumption that it never changes. This is particularly a problem when the authors use the same distribution width after major eruptions.

(b) The distribution width in the atmosphere is not static. It changes with season, altitude, latitude. The width is highly variable even when the atmosphere is not substantially impacted by volcanic and/or pyroCb activity (see Fig. 1). While the assumed width of 1.6 may be reasonable, it cannot be assumed to be static.

(c) The University of Wyoming dataset reports an uncertainty in distribution width of  $\pm 20\%$ . Therefore, even if Deshler et al. 2003 said that the width is consistently 1.6 that would still leave a range between 1.56 and 1.63. If we make some assumptions, we can model the expected behavior: assume sigma error is fixed at 20% (per Deshler et al. 2003) and the measurement error (propagated uncertainty in the extinction ratio) is only 5% (this is conservative as Taha et al. 2021 report accuracy/precision on the order of  $\pm 20\%$ ). Here (Fig. 2) we see the range of mode radii that produce extinction ratios that fall within these uncertainty bounds (everything from  $\approx$ 40 nm through 190 nm). The question then is "Which mode radius is the ' real' one, or which do you pick?" Each radius is a viable solution so the uncertainty in the authors' estimate is far larger than they show. If the authors were to use a realistic uncertainty in their estimate of distribution width (e.g., along the lines of the atmospheric variability shown in Fig. 1) and were to account for the propagated measurement uncertainty then they would see the solution space expand quite rapidly. This point cannot be overstated: the distribution width is highly important and is far from static, fixing the width to 1.6 (or any other value) imposes an unjustified constraint on the solution space and introduces bias in the inferred radius estimates as well as the corresponding number density estimate. It is for this reason that I see the method as fundamentally flawed.

The model I present in Fig. 2 is overly conservative and presents a best-case scenario. The point I'm getting to is: Even under these best-case scenarios we cannot make a definitive statement about the particle size. The requisite information content is not there.

R: All remote sensing systems make assumptions about size distributions as noted above. The actual question is: how sensitive are our results to assumptions about the size distribution in the retrieval. To address this issue, we have evaluated the impact on the retrieved size by varying distribution widths along with the color ratio. Then using the estimates of the uncertainty in color ratio and the uncertainty in distribution width, we can estimate the uncertainty in size. The results are shown in the figures below, and these figures will be added to the revised manuscript.

Figure 1 shows how the size varies with color ratio and width of the distribution derived from the Saskatran model. Given a color ratio (CR) of 3 the size varies from 0.05 to  $0.3\mu$ m over a distribution width from 1.1 to 1.8. Given a measured extinction this size range will produce a large change in the estimated aerosol concentration. However, Rieger et al. (2018) Fig 6 shows that not all distribution widths are likely, and 1.6 is a reasonable choice. Can we constrain the distribution width further, or estimate the propagation of uncertainty in the distribution width and the uncertainty in the color ratio into an uncertainty in size?

Figure 1: The size as a function of color ratio (510 nm/869 nm) and assumed particle distribution width. Color contours are  $log_{10}$  of size, black contours are size in nm. Ellipse shows an example of the domain for an uncertainty calculation for a width of 1.6 and color ratio of 3. The uncertainty is the standard deviation of the sizes within the domain.

We define the size uncertainty as the standard deviation of particle sizes within the uncertainty domain of both the color ratio (CR) and the distribution width (W). The uncertainty in CR can be estimated from Taha et al. (2021), Fig. 4. For example, the radiance uncertainty at 20 km for 879 nm is about 5% and the uncertainty at 510 nm is about 20% (both at the equator). The uncertainty in the color ratio is then CRu = sqrt ( $u_{510}^2 + u_{879}^2$ ) or 21%. For the uncertainty in distribution width for small particles, we use Fig. 6 from Rieger et al. (2018) which gives a width uncertainty (Wu) of ~0.2. To estimate the size uncertainty, we find the standard deviation for all the points within the domain W±0.2 and CR ±21% for the color ratio and the width. To get the normalized uncertainty we divide by the mean particle size within the domain. We now repeat this calculation for each CR and W value in Fig. 1. Figure 2 shows the normalized size uncertainty with contours with size overlaid.

Now we can vary the color ratio uncertainty for ranges from 18 to 28 km (radiance uncertainty up to 10%-50% for 510nm and 5%-20% for 879nm) and the distribution width uncertainty from 0.2 to 0.4. For a size value of 1.6, and averaging color ratios between 2 and 4, we find that our size uncertainty is 20%.

---

## Author Comment (AC2)

Below please find our specific responses to the reviewer RC1. Overall, the reviewer comments were quite useful and pointed to issues that need to be clarified in our analysis. Below we respond to general and specific comments. We also include a few comments from CC2 where relevant.

We would like to submit a revised paper with the additional analysis as shown below.

The format is the reviewer comments in italics followed by our reply.

**(RC1 general comments part 1):**

In this manuscript, a simple algorithm suggested for the OSIRIS instrument by Bourassa et al. [1] is applied to OMPS-LP measurements to retrieve the median radius of the log-normal particle size distribution and the number density of the stratospheric aerosols. Authors fix the mode width of the particle size distribution to 1.6 claiming this value to be typical for stratospheric aerosols. To justify this claim authors cite papers of Deshler et al. and Bourassa et al. [3, 2]. In the former paper, the only information on the mode width is given in their Fig. 5 for two single measurements. For the fine mode, it reads 1.26 in the left panel and 1.63 in the right panel of the plot. The second cited paper deals with the merging of ozone data and does not contain any information on the aerosol particle size distribution width. Thus, the claim that the width of 1.6 represents a typical value for the stratospheric aerosols remains absolutely unjustified.

The reviewer may find it beneficial to consider Rieger et al. (2018), where it is demonstrated that a width variation of 1.6 (or slightly smaller) can be considered reasonable for a background simulation. Of course, we might expect wider distributions for recent volcanic eruptions as discussed below although Bernath et al. (2023) used a narrow distribution in their analysis.

Although it is widely known that changes in the distribution width affect the extinction coefficient resulting from the Mie code in a very similar way as changes in the median radius, authors make no attempt in a course of their paper to investigate how a different assumed value of the distribution width would affect their results.

A similar comment was made by CC2 in the main text and in CC2 comment #7 shown below.

CC2: The methodology itself is fundamentally flawed and the derived products are wholly unreliable. I present a simple model below to demonstrate this unreliability. The authors assumed that the information content of 1 extinction ratio is sufficient to derive a valid estimate of particle size, but this only holds true if the distribution width is fixed and the measurement error is sufficiently small; both are invalid assumptions. While an assumed distribution width of 1.6 is a good estimate, fixing the width to that value (or any other value) imposes an artificial constraint on the solution space and inevitably biases the inferred radii and number density results. Ultimately, we have to recognize that we know very little about the atmosphere (the width could be 1.2, or it could be 1.9; both are very realistic) and forcing the distribution width to 1 specific value is wrong. Below is our response to both RC1 and CC2 comments.

We concur with your assertion that the distribution width may vary, but numerous in situ measurements have constrained the range of widths and models of the size distribution for ambient condition (Rieger et al., 2018). However, we disagree that the methodology is fundamentally flawed. In the context of this retrieval method, assuming a fixed distribution width is a necessary step and a common approach used in current retrieval algorithms, and, as the CC2 reviewer notes, 1.6 is a good estimate for the distribution width (in disagreement with R1).

The basic concern is that we have not assessed the error in assuming a fixed distribution width. We have performed a series of experiments using Mie code inside SASKTRAN to determine the sensitivity of the size to both color ratio and assumed distribution width.

Figure 1 shows how the size varies with color ratio and width of the distribution derived from the SASKTRAN model. Given a color ratio (CR) of 3 the size varies from 0.05 to  $0.3\mu$ m over a distribution width from 1.1 to 1.8. Given a measured extinction, this size range will produce a large change in the estimated aerosol concentration. However, Rieger et al. (2018) Fig 6 shows that not all distribution widths are likely. Can we constrain the distribution width further, or estimate the propagation of uncertainty in the distribution width and the uncertainty in the color ratio into an uncertainty in size?

Figure 1: The size as a function of color ratio (510 nm/869 nm) and particle distribution width. Color contours are log10 of size. Black contours are size in nm.

To do this, we define the size uncertainty as the standard deviation of particle sizes within the uncertainty domain of both the color ratio (CR) and the distribution width (W). The uncertainty in CR can be estimated from Taha et al. (2021, Fig. 4). For example, the radiance uncertainty at 20 km for 869 nm is about 5% (u869) and the uncertainty at 510 nm is about 20% (u510) - both at the equator. The uncertainty in the color ratio is then CRu = sqrt (u5102 +u8692) or 21%. For the uncertainty in distribution width for small particles, we use Fig. 6 from Rieger et al. (2018) which gives a width uncertainty (Wu) of ~0.2. To estimate the size uncertainty, we find the standard deviation for all the points within the domain W±0.2 and CR ±21% for the color ratio and the width – see ellipse in the figure. To get the normalized uncertainty we divide by the mean particle size within the domain. We now repeat this calculation for each CR and W value in Fig. 1. Figure 2 shows the normalized size uncertainty with contours of size overlaid.

---

## Author Comment (AC3)

Below please find our specific responses to the reviewer RC2. The format is the reviewer comments in italics followed by our reply. Because of the similar remarks in CC2 we have included references to CC2 below. We also responded to CC2 earlier.

*RC2:*
Overall, we appreciate the reviewer's focus on manuscript improvement.

*GC1) Finding re: number density independence (text interpreting Figure 1, lines 88-89). The first of the stated findings in the Abstract (3rd sentence), reached from interpreting Figures 1a and 1b, is not sufficiently demonstrated. The text on lines 88-89 states "This figure shows that the particle size is only a function of CR, and is independent of*
*the number density." There are 2 stated findings in the sentence about particle size, and both are questionable, unless clarified to a specific context.*

*The upper Figure (1a) shows the curve in color ratio with particle size from Mie calculations, essentially presenting how much larger the aerosol extinction is at the shorter of the two wavelengths, compared to the longer wavelength, comparing 510nm & 745nm aerosol extinction to that at the reference wavelength of 869nm. The lower Figure (1b) shows how a set of assumed number concentrations translate into aerosol extinctions at 510nm and 869nm wavelengths, for a range of assumed median sizes.*

*The reasoning for why the Figure shows this shows one can conclude the number concentration is independent of the number density is far from clear. The methodology in the paper assumes this to be the case, within a particular range of particle sizes (e.g. particle sizes sufficiently scattering at the corresponding wavelength[e.g. above some threshold value in extinction-cross-section at that wavelength]). But the text is not correct to state that can be inferred from what is shown in the Figure. I suggest to delete that text on lines 88-89, and re-write the 3rd sentence of the Abstract that states this to be a finding of the study (lines 10-11).*

Thanks for your suggestions. We will correct the text as the reviewer suggests.

*GC2) Statements re: methods too general or unclarified The sentence from GC1) is an example of several statements within the manuscript (including within the Abstract) where results are stated too generally, with insufficient communication*
*of the specifics. Given that this manuscript is within a specialist journal such as Atmospheric Measurement Techniques, the scientific writing on the methods needs to be quite precise. Whilst I understand that the text describing equation 1 is aiming to present the basis of the Bourassa et al. (2007,2008) method, the explanation on lines 75-83 need to be improved. For example the sentence on line 76 states "In computing the color ratio of aerosol extinction, the number density cancels out". Whilst that could be OK within a paragraph describing a methdological description, here this appears more prominently, and out of that context. My suggestion here is simply to delete this, expression, since it is part of the methodology already described comprehensively in Bourassa et al. (2007,2008). See specific revisions SR9 and SR10.*

>See response to SR 9, 10 below

*GC3) Comparisons to balloon-borne laser particle counter measurements (Section 3.1). This is the other part of the text where the method needs to be better explained (given this is submitted for an Atmospheric Measurement Techniques paper). The text on lines 125-126 need to provide the location of the sounding compared to, and the specific size-cut for the particle number shown in the black line in Figure 3b. (this information to be re-stated also in the Figure caption). The terminology can be confusing because the Wyoming laser-OPC (WL-OPC) was developed at Boulder (see Ward et al. 2014) and the new lightweight OPC system is called L-OPC (see Kalnajs and Deshler, 2022). The cavity-laser OPC is described in Ward et al. (2014), with multiple size channels, down to 75nm radius (75, 150, 250, 500nm, and 1.0, 2.5, 5.0, and 15.0 microns). For these comparisons to the OMPS-LP aerosol extinction, I am assuming the 75nm radius channel is shown, but this is important considering also that the original OPC40 and OPC25 only measured to 150nm particle radius (see Deshler et al., 2019). Please add, within the text on lines 125-126, and the caption to Figure 3, the minimum particle size for the size-resolved number concentration shown. Given the Mie scattering curves will of course vary for the different wavelengths considered, the minimum size is an important issue here. Related to this a suggestion is to add a dashed line for the R>150nm number concentration (and possibly also the 250nm line, in dot-dashed or so).*

We agree with these excellent suggestions. We modified the text as suggested.

*The overestimation shown in the 16-18km altitude-range could potentially be due to only some proportion of those R>75nm particles being measured, even at the shorter of the two OMPS-LP wavelengths. I appreciate this is a retrieval, but then the issue of what particle sizes are represented within the two aerosol extinction metrics within the color-ratio particle size method probably justifies considering an uncertainty-range across more than 1 of the size channels (particle size cuts) in the comparisons.*

We added an uncertainty calculation section in the revised manuscript as follows:

Adjusting the size distribution will affect the cross section and the Mie phase function. To investigate the OMPS-LP aerosol color ratio sensitivity to the assumed phase function, we perturb the phase function parameters by ± 10% and run the OMPS-LP retrieval algorithm at a range of scattering angles observed during a single orbit, similar to Chen et al. (2018). OMPS-LP V2.0 aerosol retrieval algorithm assumes a gamma aerosol size distribution using fitted parameters of $\alpha = 1.8$ and $\beta = 20.5$ where effective radius reff = 0.185μm. Their conversion formular (Chen et al., 2018) is as follows

$$r_{eff} = \frac{(\alpha + 2)}{\beta} \qquad (5)$$

which is used to calculate the aerosol phase function (see Fig 1). For a width W=1.6, we can get a modal radius rm = 0.1μm by the following formular,

$$r_{eff} = r_m \exp\left(\frac{5}{2} ln^2 S\right) \qquad (6)$$

Figure 2 shows that the phase function is more sensitive to β changes than α. A 10 % change of β can produce ± 10 and 15% change in 510 nm and 869 nm phase functions, respectively, while a 10 % change of α results in a ± 3 and 5% change for both wavelengths. Chen et al. (2018) noted that increasing α increases the peak of the differential size distribution while increasing β shifts

the peak distribution to a larger particle radius. Figure 2b shows that the phase function perturbations produce anti-correlated but lesser changes in aerosol extinction. The changes caused by β perturbations are mostly within 5% for both wavelengths and 3% for α. The structural change in the extinction coefficient along the orbit is caused by scene reflectivity (Chen et al., 2018). The effect on extinction ratio (510/869) is shown in Fig. 2c for the same variations of phase functions. The color ratio perturbations are mostly within 3% between scattering angles of 65-125° and 5% outside that range, except for the very small scattering angles, which might be caused by scene reflectivity changes. Let's pick a typical color ratio of 3 with W = 1.6 to calculate the uncertainty of retrieved particle size. The size range would be 0.096 ~ 0.103 µm for 3% change in color ratio perturbations and 0.094 ~ 0.106 µm for 5% change in color ratio perturbations. It means the 3% and 5% change in color ratio perturbations lead to < 4% and < 7% difference in retrieved aerosol particle size, respectively.

[Figure]

Figure 1: Plot of the aerosol phase function used for OMPS LP retrieval algorithm, which assumes gamma size distribution at 510 (blue) and 869 nm (red). Rayleigh phase function is also shown (black).

[Figure]

Figure 2: plot of simulated phase function (a), aerosol extinction (b), and color ratio (c) perturbations as caused by gamma parameter changes of ± 10%. Perturbations are shown relative to OMPS LP operational retrieval at various scattering angles. Scattering angles represent the range of are in the northern hemisphere, while large scattering angles are in the southern hemisphere (Taha et al., 2021). The aerosol extinction was retrieved using a single orbit on 12 September 2016 at 20.5 km altitude.

*List of specific revisions*
*--------------------------*
*SR1) Abstract, lines 8-9 -- This initial text "We have developed an algorithm" might make some readers assume the MS is to explain some development of a new algorithm, but the MS is rather applying an existing method already developed for the OSIRIS data (Bourassa et al., 2007, 2008), to OMPS measurements, comparing to in-situ measurements for two recent volcanic case studies. For this initial sentence better to put the object of the sentence (derive the size...) at the start of the sentence, with also referring more generally to "aerosol microphysical parameters" rather than the specifics in this initial sentence.*
*Please change "We develop an algorithm that uses the aerosol extinction at two wavelengths..." instead to "We apply an existing method to derive aerosol microphysical parameters from OMPS dual-wavelength aerosol exinction measurements, to analyse particle size variations*

*for two recent stratospheric volcanic case studies." or similar.*

Good point. We will modify the text as suggested.

*SR2) Abstract, lines 9-10*
*With the above re-wording, the first part of the 2nd sentence of the Abstract is already integrated into the above re-worded 1st sentence, and the latter part re: SAGE should be stated later, being clear this is SAGE-III on ISS.*

We will correct.

*SR3) Abstract, line 10 -- This 3rd sentence of the Abstract also needs to be re-worded, with the current text "We show that the color ratio between two wavelengths is insensitive to number density." suggesting the manuscript is presenting this as a finding from the study.*
*Figure 1 is very useful in showing the variation with particle size and relative magnitudes of monochromatic aerosol extinction at 3 specified wavelengths, for a log-normally distributed liquid aqueous sulfuric acid solution aerosol population (for an assumed refractive index spectrum and water content/weight-percent). However, as is explained in General Comment 1 above, the text on lines 88-89 is not correct in stating the Figure shows the color ratio is independent of number density. I understand this is an assumption within the methods explained in the Bourassa et al. (2007, 2008) studies, and for a particular range of particle sizes, it's a reasonable assumption to make. However, the Abstract should not present as a finding from the study. Also, the "and thus" in the 2nd part of the sentence (with the current wording) does not follow, and suggests a misunderstanding re: what is assumed in the method, with the wording "We show the color ratio between two wavelengths (e.g. 510nm/869nm) is insensitive to aerosol concentration, and thus can be used to derive aerosol size assuming a log-normal size distribution."*

*Within the re-constructed sentence, the color ratio metric also needs to better communicated, a suggested re-wording to have "of aerosol extinctions at" in place of "between", and change "(e.g. 510nm/869nm)" instead to "(510mm and 869nm)". My specific suggestion for the 2nd part of this sentence is to delete "and thus can be used to", and also change "We show that" instead to a re-worded sentence similar to below: "The color ratio between aerosol extinction at two wavelengths is used to derive from satellite measurements large-scale particle size variations within volcanic aerosol clouds in the stratosphere."*

We will make these changes.

*SR4) Abstract, lines 12 -- Re-word "With the size and extinction, we can compute a number density consistent with both wavelengths". The particle size is being inferred from the two-wavelength aerosol extinctions, not measured directly. With the re-constructed preceding sentences (from SR3), suggest to instead have this sentence be to explain some greater specifics in the method, prior to its application for profiling the size distribution of the Raikoke and Hunga Tonga aerosol clouds. A suggested re-wording "With the size and extinction, we can compute..." instead with "As a further microphysical Consistent with the two extinction*

Good suggestions, we will implement these.

*SR5) Abstract, lines 13-14 -- Please re-word this sentence to be more specific re: the Comparisons between the particle size variations from OMPS-LP and those derived from SAGE-III on ISS, giving some specifics about how well the two size products compare.*

It is difficult to describe these comparisons in a single sentence, but we will modify the sentence to read: Our size and concentration estimates are also in good agreement between 24 and 18 km with measurements made by the Stratospheric Aerosol and Gas Experiment on the International Space Station (SAGE III/ISS) over multiple scattering angle ranges.

*SR6) Introduction, lines 21-22 -- change "have been connected to short-term changes in climate" instead to "major eruptions causing substantial short-term changes in climate", or similar. The first sentence (of this 1st paragraph of the introduction) mentions the volcanic impacts on climate, but this adaptation of the sentence is then consistent with the strong (but temporary) radiative forcings after historical very large-magnitude explosive tropical eruptions.*

We agree and clarify this.

*SR7) Introduction, line 25 -- Please re-word this sentence to be clearer what is meant by "dust" in this context ("Volcanic ash, Pyro-CB smoke and dust") -- presumably it is cosmic dust (i.e. meteoric aerosol) that is meant here, right? Suggest to have the re-worded text provide this information in relation to citing a recent observational studies for each of these non-sulphate stratospheric aerosol constituents. Suggestions are Vernier et al. (2016), for volcanic ash, Khaykin et al. (2020) for pyro-Cb smoke, and Schneider et al. (2021) for meteoric aerosol. With this re-wording the text on line 26, suggest to cite paper also for the volcanic ash Heating (Muser et al., 2020), and cite the Yu et al. (2019) for the wildfire smoke heating effect. The cosmic dust (meteoric smoke) may be present at sizes smaller than for wildfire smoke and volcanic ash, also some components dissolving into the sulphuric acid solution aerosol particles (see James et al., 2023), the heating effect is primarily associated with the smoke and ash.*

No, we meant mineral dust which has been observed in the lower stratosphere by research aircraft. We agree that references would be helpful here and will include them in the revision.

*SR8) Introduction, lines 32-34*
*Re-word "has provided global monitoring of the stratospheric aerosol layer since 1975" because the SAGE and SAM-II record only began in 1979 (see McCormick et al., 1977). The original testing of the SAM sensor on the Apollo Soyuz mission was in 1975, but the global monitoring only began in 1979. Re: the text on the solar occultation methods, the so-called "onion-peeling" retrieval method was originally developed in the 1960s (Edward Ney group at the University of Minnesota), and it will help retain the heritage of the instrument development to cite the Pepin (1969) report that first documented the technique, and the Rosen et al. (1969) which shows (Figure 4) the only peer-reviewed paper showing the initial application of the methods to balloon-borne solar extinction measurements, and later further developed for application to the*

*first satellite measurements of the stratospheric aerosol layer (the SAM, SAM-II, McCormick et al., 1979), and then SAGE, SAGE-II and SAGE-III series of satellite measurements.*

Nice history, CC1 also provided some references on the technique that we will include in the revised manuscript.

*SR9) Section 2, line 76-78*
*Add ", at wavelength l, " after "The aerosol extinction" and add subscript*
*l symbol to the two wavelength-dependent symbols in equation 1 (sigma and AR).*
*In addition, please change the abbreviation "AE" for aerosol extinction, as this may*
*initially confuse some readers, given the related size-associated aerosol metric*
*"Angstrom exponent". Please change all occurrences of "AE" instead to either "k"*
*(used in the Bourassa et al. (2007, 2008) papers) or other terminology*
*(the letter b is used, subscript "ext" to denote aerosol extinction in*
*Seinfeld & Pandis (2006). It also needs to be stated explicitly that these calculations are based*
*on an assumed log-normal size distirbution, with also the associated water content (sulphuric*
*acid and water composition). As explained in General Comment GC2, please delete the two*
*short sentences lines 77-78 as this statement does not follow from what is shown. The*
*information and associated assumptions are explained within context in the Bourassa et al.*
*(2007,2008) papers.*

We will implement these suggestions.

*SR10) Section 2, lines 82-83*
*As mentioned in General Comment GC2, the text "...but as the two wavelengths approach each other" is too colloquial and needs to be formalised. Also, the "Any two distinct wavelengths can be chosen" at the start of the sentence also does not need to be stated. Suggested restructuring to better explain the issue. Suggest to re-word the original text: "Any two distinct wavelengths can be chosen for CR, but as the two wavelengths approach each Other Fig. 1a shows that the CR gradient decreases and thus the uncertainty in the retrieved size increases." instead to "For OMPS-LP wavelengths closer to the reference 869nm wavelength, the gradient in the colour ratio curve is less steep, causing an increased uncertainty in the retrieved particle size." It is worth noting also that the 869nm aerosol extinction measurement is "cleanest" retrieval in relation to effects from other species, and this must also be considered in relation to how the errors/uncertainty differ between different color-ratio wavelength-pairs.*

We will implement these suggestions.

*SR11) Figure 1 caption, lines 431-435. Although it is not stated explicitly, my understanding is that these calculations are based on a log-normally distributed aerosol, with geometric standard deviation of 1.6, following the specification within SASKTRAN. This should be stated in the caption to Figure 1.*

Yes

*SR12) Section 2, line 368 -- formatting error re: Loughman et al. (2015) reference.*

Corrected. Thanks.

*References*
*----------*
*Bourassa et al. (2007) Stratospheric aerosol retrieval with OSIRIS limb scatter measurements,*
*J. Geophys. Res., 112, D10 217, https://doi.org/10.1029/2006JD008079.*

*Bourassa et al. (2008) Retrieval of stratospheric aerosol size information from OSIRIS*
*limb scattered sunlight spectra, Atmos. Chem. Phys., 8, 6375‚Äì6380,*
*https://doi.org/10.5194/acp-8-6735-2008.*
*James et al. (2023) The importance of acid-processed meteoric smoke relative to meteoric*
*fragments for crystal nucleation in polar stratospheric clouds*
*Atmos. Chem. Phys., 23, 2215‚Äì2233, https://doi.org/10.5194/acp-23-2215-2023*

*Khaykin et al. (2021) "The 2019/20 Australian wildfires generated*
*a persistent smoke-charged vortex rising up to 35 km altitude"*
*Comms. Earth Env., https://doi.org/10.1038/s43247-020-00022-5*

*McCormick et al. (1978) Satellite studies of the stratospheric aerosol*
*Bulletin of Amer. Meteorol. Soc., vol. 60, no. 9, 1038-1046.*
*https://doi.org/10.1175/1520-0477(1979)060<1038:SSOTSA>2.0.CO;2*

*Muser et al. (2020) Particle aging and aerosol‚Äìradiation interaction affect*
*volcanic plume dispersion: evidence from the Raikoke 2019 eruption,*
*Atmos. Chem. Phys., 20, 15015‚Äì15036, https://doi.org/10.5194/acp-20-15015-2020*

*Pepin T. J. (1969) "The Use of Extinction from High Altitude Balloons as a Probe*
*of the Atmospheric Aerosols", Univ. Minn. progress report, Oct, 1969*
*https://apps.dtic.mil/sti/pdfs/AD0696527.pdf*

*Pepin, T. J. (1977) "Inversion of solar extinction data from the Apollo-Soyez test project*
*Stratospheric aerosol measurement (ASTP/SAM) experiment", volume 1, 529-544, NASA report*
*SP-442. https://ntrs.nasa.gov/citations/19780009145*

*Rosen, J. M. (1969) "Stratospheric dust and its relationship to the meteoric influx",*
*Space Science Reviews, vol. 9, 58-89, https://doi.org/10.1007/BF00187579 (see Figure 4).*

*Schneider et al. (2021) " Aircraft-based observation of meteoric material in*
*lower-stratospheric aerosol particles between 15N and 68N", Atmos. Chem. Phys.,*
*vol. 21 989-1013, https://doi.org/10.5194/acp-21-989-2021*

*Seinfeld and Pandis (2006) Atmospheric Chemistry and Physics: From Air Pollution to Climate*
*Change, 2nd Edition, Wiley publisher.*

*John H. Seinfeld, Spyros N. Pandis Vernier et al. (2016) "In situ and space-based observations of the Kelud volcanic plume: The persistence of ash in the lower stratosphere", J. Geophys. Res., vol. 121, 11,104‚Äì11,118, [http://doi.org/10.1002/2016JD025344](http://doi.org/10.1002/2016JD025344).*

Reference:

Chen, Z., Bhartia, P. K., Loughman, R., Colarco, P., and DeLand, M.: Improvement of stratospheric aerosol extinction retrieval from OMPS/LP using a new aerosol model, Atmos. Meas. Tech., 11, 6495–6509, https://doi.org/10.5194/amt-11-6495-2018, 2018.

---

## Author Comment (AC4)

Below please find our specific responses to the reviewer CC1. The format is the reviewer comments in italics followed by our reply.

CC1:

1) It is "Raikoke", not "Reikoke" Yes, sorry.

2) L25: "dust": do you rather mean "meteoritic dust"?

No, we meant mineral dust which has been observed in the lower stratosphere by research aircraft.

3) L78-79: "Figure 1a shows...869 nm": what are your assumptions in terms of composition in these Mie calculations (I guess it is sulphates but in case please mention this explicitly"

We assumed spherical sulfate aerosol particles with usual refractive index. We will add that point to the text.

4) Linked to my previous question, one fundamental question that I have on this method: how do you cope in your method with aerosol mixing, i.e. aerosol layers with particles of different composition? In this case, I imagine that CR is no more insensitive to size. This is quite critical for first stages of some volcanic eruptions, like Raikoke (there was a significant fraction of ash in the early sulphate plume).

The short answer is that we don't assume aerosol mixing, the different sizes are accounted for In the size distribution which is assumed fixed (log-normal in the case of OMPS-LP). However, our error analysis (above) quantifies the uncertainty in the retrieved size as a function of the size distribution width.

5) For Hunga Tonga, there are size distribution measurements shown in Kloss et al. 2022 (https://agupubs.onlinelibrary.wiley.com/doi/10.1029/2022GL099394), that are ideally perfect correlative data for your method.

Yes, thank you for the reference.

6) For the rapid formation of sulphate aerosol in Hunga Tonga plume (which is linked to their size evolution), you cite model studies of Zhu et al 2022 (L234) but this is also shown with observations in Sellitto et al. 2022

(https://www.nature.com/articles/s43247-022-00618-z), as well as hypothesis of why observed SO2 emissions where small (L228) and estimations of the radiative impacts of Hunga Tonga plume (L227), and should then be cited in your discussion.

We will include the Silletto et al. (2022) reference in the revised paper.

---

## Author Comment (AC5)

Below please find our specific responses to the reviewer CC3. The format is the reviewer comments in italics followed by our reply.

*CC3:*

*1. The interpretation of the color ratio depends entirely on Mie scattering calculations carried out with SASKTRAN. These calculations depend on input parameters and assumptions, which need to be explicitly stated. In particular, the results depend on the assumed composition (weight % of sulfuric acid) and the optical constants used. What is the temperature of the optical constants and was there any effort to match the optical constant temperature to the atmospheric temperature? The assumption of 1.6 for the width of the log-normal distribution is noted, but the systematic error on the median radius from this assumption is not estimated. Some discussion of systematic errors introduced by these various assumptions should be included.*

Thanks for the suggestions, which is similar to RC1. We will include error estimates in the revised manuscript as Sections 2.2 and 2.3.

*2. Two recent papers on the properties of stratospheric sulfate aerosols from Raikoke, Tonga and Nabro volcanic eruptions based on ACE-FTS spectra have been overlooked [1,2]. These papers can help with point 1 above and with particle size comparisons with independent measurements.*

*References*
*1. Bernath, C. Boone, A. Pastorek, D. Cameron and M. Lecours, Satellite characterization of global stratospheric sulfate aerosols released by Tonga volcano, J.*
*Quant. Spectrosc. Rad. Transfer 299, 108520 (2023). Doi: 10.1016/j.jqsrt.2023.108520*
*2. D. Boone, P. F. Bernath, K. LaBelle and J. Crouse, Stratospheric Aerosol Composition Observed by the Atmospheric Chemistry Experiment Following the 2019*
*Raikoke Eruption, J. Geophys. Res.: Atmospheres 127, e2022JD036600 (2022). Doi: 10.1029/2022JD036600*
*3. D. Cameron, P. Bernath and C. Boone, Sulfur Dioxide from the Atmospheric Chemistry Experiment (ACE) Satellite, J. Quant. Spectrosc. Rad. Transfer 258,*
*107341 (2020). DOI: 10.1016/j.jqsrt.2020.107341*

Thank you for these references, we will include them.

*3. The plume from the Raikoke volcanic eruption traveled both northwards and southwards, not just South [2,3]. "The eruption cloud is initially at 50° N and moves southward so the aerosols are detected at more southerly latitudes at a later time."*

Yes, a portion of the plume moved northward, but in the figures we see a time phasing as it moved southward. Figure 8 from Gorkavyi et al. (2021) clearly shows the spread in both directions. We see how the sentence could be confusing and will correct it.

Reference:

Gorkavyi, N., Krotkov, N., Li, C., Lait, L., Colarco, P., Carn, S., DeLand, M., Newman, P., Schoeberl, M., Taha, G., Torres, O., Vasilkov, A., and Joiner, J.: Tracking aerosols and SO2 clouds from the Raikoke eruption: 3D view from satellite observations, Atmos. Meas. Tech., 14, 7545–7563, https://doi.org/10.5194/amt-14-7545-2021, 2021.